# Charged biexciton polaritons sustaining strong nonlinearity in 2D semiconductor-based nanocavities

Ke Wei[1,3] ✉, Qirui Liu[2,3], Yuxiang Tang[1], Yingqian Ye[2], Zhongjie Xu[2] & Tian Jiang ●[1] ✉

Controlling the interaction between light and matter at micro- and nano-scale can provide new opportunities for modern optics and optoelectronics. An archetypical example is polariton, a half-light-half-matter quasi particle inheriting simultaneously the robust coherence of light and the strong interaction of matter, which plays an important role in many exotic phenomena. Here, we open up a new kind of cooperative coupling between plasmon and different excitonic complexes in $WS_2$-silver nanocavities, namely plasmon-exciton-trion-charged biexciton four coupling states. Thanks to the large Bohr radius of up to 5 nm, the charged biexciton polariton exhibits strong saturation nonlinearity, ~30 times higher than the neutral exciton polariton. Transient absorption dynamics further reveal the ultrafast many-body interaction nature, with a timescale of <100 fs. The demonstration of biexciton polariton here combines high nonlinearity, simple processing and strong scalability, permitting access for future energy-efficient optical switching and information processing.

The quest for condensed-matter states with effective optical response and giant optical nonlinearity has been long sought in the modern physics community[1–3]. A strongly correlated electron–photon system[4], like coherent exciton–polariton in strong light–matter coupling regime[5–7], is becoming one of the most hopeful candidates in this research strand owing to its mixed nature. By virtue of the photonic component, exciton–polariton was intensively investigated in the phenomenon of Bose–Einstein condensation and demonstrated threshold-less lasing[8–11] as well as long-range fluid propagation up to tens of micrometers[12,13]. While relying on the matter component, exciton–polariton presents strong interparticle interactions, leading to the emergence of rich nonlinear behaviors such as parametric scattering[14] and antibunching of nonclassical light[15]. Atomically thin transition metal dichalcogenides (TMDs) hosting Wannier–Mott exciton with large binding energy and oscillator strength have recently gained much attention in the development of polariton devices[14,16],

due to their robust room-temperature stability and enhanced Coulomb interactions beyond the hydrogenic picture[17]. Nevertheless, polariton nonlinearity in TMDs is still not strong enough to trigger the related device applications due to the intrinsically small Bohr radius of the exciton. One popular proposal to address this issue is to form polaritons with higher-order excitations that have stronger interactions. Examples including highly nonlinear trion-polaritons[18–20], Rydberg exciton-polaritons[21,22], moiré exciton-polaritons[23], polaron-polaritons[24], and dipolar exciton-polaritons[25,26] have been successfully illustrated for now, with the study of biexciton–polaritons remaining elusive.

Although the formation of polaritons with biexciton is a promising way to boost the nonlinear interaction by taking advantage of its large Bohr radius[21,23–27], it is still a great challenge to achieve these coherent phenomena due to the intrinsically low oscillator strength of the four-particle excitonic state and diffraction-limited

[1]Institute for Quantum Science and Technology, College of Science, National University of Defense Technology, 410073 Changsha, China. [2]College of Advanced Interdisciplinary Studies, National University of Defense Technology, 410073 Changsha, China. [3]These authors contributed equally: Ke Wei, Qirui Liu. ✉e-mail: weikeaep@163.com; tjiang@nudt.edu.cn

optical mode volume of the prototypical dielectric microcavity. A very recent report[28], however, shows that a five-particle Coulomb correlation state−charged biexciton ($XX^-$) in monolayer $WS_2$, displays an anomalously high light absorption comparable to that of trion (T) and neutral exciton (X), arising from the doping effect of existing donors. On the other hand, for the cavity photon component, localized plasmonic resonances based on metallic small-scale cavities are frequently utilized to obtain strong light−matter coupling strength in TMDs due to their deep subwavelength mode confinement[29–33]. Therefore, the combination of $XX^-$ with plasmonic nanocavity is highly expected to provide us an excitedly preceding opportunity to access the observation of biexciton−polaritons in TMDs, and thus laying a foundation for the preliminary study of biexciton−polaritons nonlinearity. Here, we establish a hybrid plasmon−X−T−$XX^-$ polariton state in heterostructures consisting of silver nanodisk arrays (Ag ND) and monolayer $WS_2$ prepared by chemical vapor deposition (Fig. 1a). Although the sample suffers severe loss and broad linewidth compared with the precisely designed microcavity and the preferred high-quality mechanical exfoliated material, the strong interaction between the plasmon and excitonic state still enables cooperative couplings. Spectral measurements demonstrate a huge saturation nonlinearity of the $XX^-$ polariton, about 30 times that of the X polariton, arising from the large Bohr radius and possible trap doping depletion effect at high pump fluence.

## Results

### Excitonic complexes and polaritons in monolayer $WS_2$-based nanocavity

To realize coherent coupling between different excitonic complexes and nanocavity plasmon, the $WS_2$−Ag ND hybrid samples were constructed on transparent fused quartz substrates by wet transfer technique, where the detuning between the exciton and

plasmon resonances is obtained by changing the ND diameter (see the "Methods" section and Supplementary Note 1 in Supplementary Information for details). Low temperature (4 K) PL spectrum (see Supplementary Fig. 2) reveals a large number of trap states in $WS_2$, which exhibits a dominant broadband emission peak below 2 eV at low and medium power. These shallow-level trap states may cause electron doping in $WS_2$ through photogating effect[34–37], and significantly enhance the oscillator strength of the charged multi-particle complexes such as T (2.064 eV) and $XX^-$ (2.040 eV)[38–40], resulting in pronounced optical absorption comparable to that of neutral X (2.092 eV) (Fig. 1b). The large oscillator strength, in conjunction with the nanoscale optical field confining of the plasmonic nanocavity, allows the three excitonic complexes to strongly couple with the plasmon generated at the $WS_2$−Ag interface (Fig. 1a). As the plasmonic linewidth is considerably larger than the energy difference of different exciton resonances, such strong coupling manifests as a hybrid plasmon−X−T−$XX^-$ polariton state, with three spectral dips coinciding the exciton resonances and four peaks representing the polariton eigenmodes, namely, the lower (LP), first middle (MP1), second middle (MP2) and upper (UP) polaritons.

To shed light on the nature of the quadruplet coupling state, reflectance contrast ($R_c$) spectra under different detunings are measured at 4 K, as shown in Fig. 1c. Four polariton eigenmodes with anti-crossing behavior are clearly observed, indicating strong or intermediate coupling between the excitons and plasmon. Similar experiments are also performed at 293 K (Supplementary Fig. 9), where only X retains due to the dissociation of T and $XX^-$. To quantitatively ascertain the coupling strength, polariton resonances are extracted versus the plasmon energy (Fig. 1d, e). In close vicinity of the exciton resonance, anti-crossing dispersions at 4 K can be captured by a four-coupled

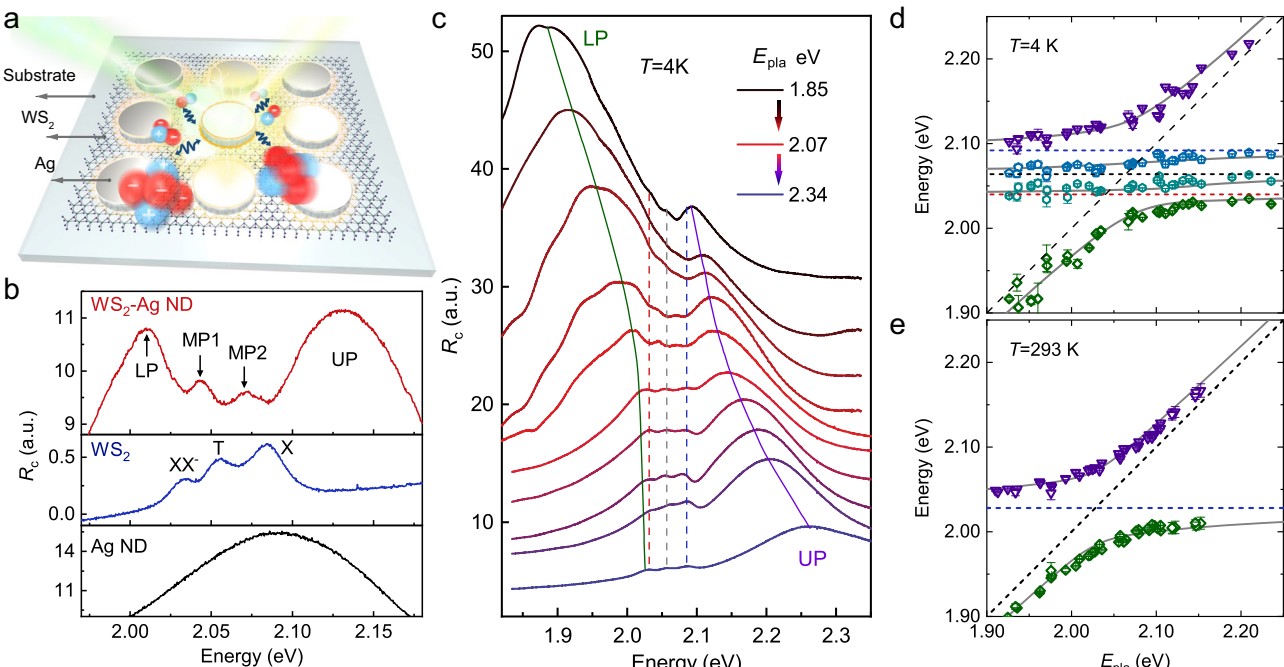

**Fig. 1 | Cooperative coupling between different excitonic complexes and plasmon in $WS_2$−Ag hybrid system. a** Schematic of the hybrid sample. The Ag NDs are periodically patterned on monolayer $WS_2$, and coherent polaritons are generated near the bottom ring of the NDs. **b** $R_c$ spectrum of $WS_2$−Ag (top), bare $WS_2$ (middle), and Ag NDs (bottom), showing the four polariton eigenstates and three excitonic resonances. **c** $R_c$ spectra with different detunings acquired by varying the

ND diameter. Clear anticrossing behavior of UP and LP (solid lines) is found, along with the three excitonic resonances guided by dashed lines. **d** and **e** Anticrossing dispersions of the polariton eigenstates at 4 K (**d**) and 293 K (**e**). Dots are extracted from experimental $R_c$ spectra shown in (**c**) and Supplementary Fig. 9a, while lines are the calculated eigenenergies. The error bars represent 95% confidence intervals.

Hamiltonian (similar two-coupled model at 293 K)

$$H = \begin{pmatrix} E_{pla} - i\gamma_{pla}/2 & \Omega_X/2 & \Omega_T/2 & \Omega_{XX^-}/2 \\ \Omega_X/2 & E_X - i\gamma_X/2 & 0 & 0 \\ \Omega_T/2 & 0 & E_T - i\gamma_T/2 & 0 \\ \Omega_{XX^-}/2 & 0 & 0 & E_{XX^-} - i\gamma_{XX^-}/2 \end{pmatrix} \quad (1)$$

where $E_{pla}$ and $\gamma_{pla}$ denote the energy and linewidth of the plasmon, $E_{X,T,XX^-}$ and $\gamma_{X,T,XX^-}$ are the energies and linewidths of the three excitonic complexes, and $\Omega_{X,T,XX^-}$ represent their coupling strengths with plasmon. Here, we assume that the direct interactions between different exciton complexes are negligible due to the ultrafast coherent lifetime of the coupling system (typically a few tens of femtoseconds)[41].

The diagonalization of the Hamiltonian at 4 K provides four fitting polariton eigenmodes matching well with the experimental results (Fig. 1d), with coupling strength of $\Omega_{X,T,XX^-} = 88, 80, 70$ meV. From independent $R_c$ spectrum measurement of the Ag ND (Supplementary Fig. 1) and pure WS$_2$ (Supplementary Fig. 3), we obtain $\gamma_{pla} = 180$ meV and $\gamma_{X,T,XX^-} = 26, 25, 18$ meV. The X linewidth here is apparently larger than the reported values at similar temperatures (typically < 10 meV)[23,30,42], possibly caused by the inhomogeneous broadening from the trap states. While the large plasmonic linewidth indicates a tremendous intrinsic energy dissipation of metal nanostructure, which inhibits the Rabi oscillation between plasmonic and excitonic modes. As a result, only a strong coupling regime is reached here for X and T, with $\Omega_{X,T} > (\gamma_{pla} - \gamma_{X,T})/2$[43]. While for XX$^-$, $\Omega_{XX^-}$ is slightly smaller, but comparable to $(\gamma_{pla} - \gamma_{XX^-})/2$ and far beyond $\gamma_{XX^-}$ suggesting a coherent hybridization at initial excitation[44] composed of mode splitting ($\Omega_{X,T} > (\gamma_{pla} - \gamma_{X,T})/2$) and Fano interference ($\Omega_{X,T} < (\gamma_{pla} - \gamma_{X,T})/2$). This coupling strength, also known as intermediate coupling, has been extensively studied in recent years[31,32], especially in plasmon–exciton hybrid system[44,45]. In this moderate regime, cooperative couplings and new polariton eigenstates have already emerged, which can also be characterized by the polariton framework. Increasing the temperature from 4 to 293 K provides $\Omega_X = 140$ meV, $\gamma_X = 40$ meV, and $\gamma_{pla} = 244$ meV (Fig. 1e), at which a strong coupling condition is also met. Note that

$\Omega_X$ at 293 K is in excellent agreement with $\sqrt{\Omega_X^2 + \Omega_T^2 + \Omega_{XX^-}^2} = 138$ meV at 4 K. Given that the coupling strength ($\Omega$) depends on the oscillator strength ($f$) within the plasmonic volume[46], i.e., $\Omega \propto \sqrt{f}$, this consistency indicates an entire distribution of the $f$ between the three exciton species during refrigeration.

To examine the thermal stability of the coupling system, the $R_c$ spectrum of the zero-detuning sample is measured under continuously varying temperatures at two selected pump fluences, as shown in Fig. 2a. Spectral dip at XX$^-$ resonance is found up to 120 K for low pump fluence, at which the $\Omega_{XX^-}$ dramatically decreases to 28 meV from fitting the spectrum. This coupling strength is far less than the intermediate coupling condition and thus is categorized into Fano interference. Despite the absence of new polariton eigenmodes, the Fano interference here points out the apparent XX$^-$ oscillator strength at a temperature of >100 K. The critical temperature drops to ~60 K at high pump fluence (Fig. 2b), hinting at a large nonlinearity of XX$^-$ polariton, which will be discussed in the next section.

## High nonlinearity of charged biexciton polaritons

Nonlinear optical responses of the cooperative couplings between plasmon and different exciton complexes are investigated via the polariton density-dependent $R_c$ spectrum of the zero-detuning sample (Fig. 3a). To precisely determine the polariton density consisting of different exciton complexes, we develop a calibration procedure in Supplementary Note 4. Here, the injecting total polariton density $n_{tot}$ ranges from $1 \times 10^3$ to $3 \times 10^5$ $\mu m^{-2}$ as the average power of the incident femtosecond white light varies from 1 to 207 nW. With increasing density, the UP and LP branches shift toward the intermediate exciton resonance, with $\Delta E_{LP} = 4.4$ meV and $\Delta E_{UP} = -6.3$ meV (dots in Fig. 3c), and the three excitonic dips also become shallower, of which the XX$^-$ dip collapses most pronouncedly, indicating strong nonlinear interaction of XX$^-$ polaritons.

To give a quantitative analysis, we fit the $R_c$ spectrum using the above four-coupled oscillator model combined with a Lorentz base, which takes into account the uncoupled pure plasmon generated near the upper ring of the Ag ND (see Supplementary Note 3 for more details). The fitting curves precisely reproduce the measured spectra at all pump fluences, with coefficients of determination of $R^2 > 0.992$ (Fig. 3b). The resonance energies of the four polariton eigenmodes extracted from the fitting curves are also in excellent agreement with that from the experimental data (Fig. 3c and Supplementary Fig. 6b). In contrast, fitting results without considering the Lorentz base fail to catch these fine features (Supplementary Fig. 5), confirming the strong light absorption from the uncoupled plasmon reservoir. This is fundamentally different from the dielectric microcavity, where light is mainly absorbed by the excitonic component[25].

The coupled model allows us to extract the intrinsic properties of the polaritons, including the exciton resonance $E$, linewidth $\gamma$, and coupling strength $\Omega$. Here, for comparison, we mainly concentrate on X and XX$^-$ polaritons, as shown in Fig. 3c–e. Similar results of T polariton can be found in Supplementary Figs. 6 and 7. With increasing density, the X resonance exhibits a negligible blueshift (<0.5 meV) within the fitting error, pinning down a minor repulsive coulombic exchange interaction for the plasmon-exciton hybrid system in the sample. We notice the band redshifts of the XX$^-$ and T at high pump density (Supplementary Fig. 6a), possibly caused by the increasing kinetic energy[47] or density[48] of the photodoped electrons, both of which may lead to increasing energy splitting between the charged multi-particles and X. As a consequence, the intrinsic saturation and broadening dominate the nonlinear behavior, which stems from phase space filling (i.e. Pauli blocking) and excitation-induced dephasing, respectively, and can be depicted well with the classical response

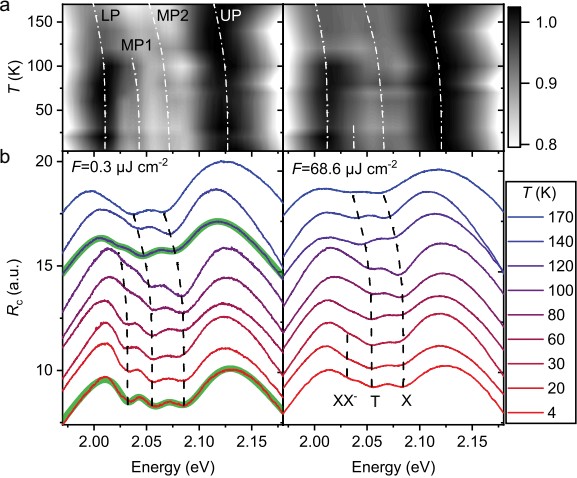

**Fig. 2 | Temperature-dependent coupling states. a** Contour plots of the $R_c$ spectra of a zero-detuned hybrid sample at two distinct pump fluences of 0.3 µJ cm$^{-2}$ (left) and 68.6 µJ cm$^{-2}$ (right). The white dash dots mark the resonances of the four polaritons, which manifest as peaks in the spectra. **b** Corresponding $R_c$ spectra at selected temperatures showing the fine structure of the coupling system. The black dashes label the excitonic energies, which manifest as dips in the spectra. The two green areas in the left panel are fitting results using the four-coupled oscillator model combined with a Lorentz base.

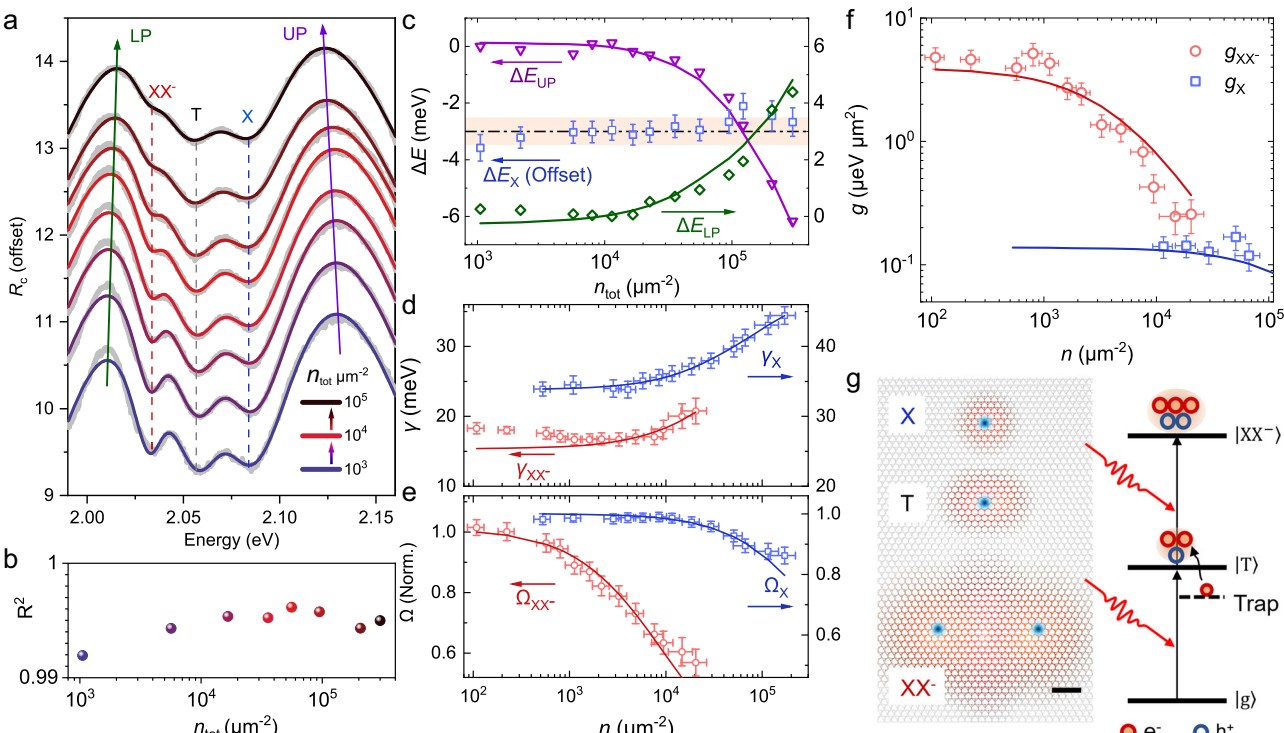

**Fig. 3 | Nonlinearity of different exciton polaritons. a** $R_c$ spectra of zero-detuned sample with increasing total polariton density $n_{tot}$ from bottom ($1 \times 10^3$ μm$^{-2}$) to up ($2 \times 10^5$ μm$^{-2}$). The experimental data (black lines) are vertically offset for clarity and overlapped by the fitting data (red lines), with the vertical dashed lines representing the exciton resonances, and the arrows guiding the LP and UP resonances. **b** Coefficient of determination ($R^2$) of the fits in (**a**). **c** Shifts of the LP, UP and X resonance versus $n_{tot}$, extracting from the experimental (dots of LP and UP) and fitting (solid lines of LP and UP, and X) data in (**a**). The shadows are guides for the eyes, showing the tiny shift of $E_X$. **d** and **e** Shifts of the linewidth (**d**) and coupling strength (**e**) of the polaritons versus their corresponding densities. Dots are extracted from the fitting curves in (**a**), and lines are fitted with Eq. (2). **f** Calculating saturation nonlinearity ($g(n) = |d\Omega(n)/dn|$) from corresponding data in (**e**). All the error bars in this figure represent 95% confidence intervals. **g** Left, schematics of wavefunction distribution of the three excitonic complexes. The blue and red areas represent the distribution of hole and electron clouds, respectively. Scale bar: 1 nm. Right, schematics of the light absorption mechanism of XX$^-$, where the WS$_2$ absorbs two photons ($\hbar\omega_{XX^-}$) and transitions from the ground state to the XX$^-$ excited state, with the aid of the energetic electron ionized from the shallow trap states (see Supplementary Note 2).

model for 2D excitons[32,49,50]:

$$\Omega(n) = \Omega(0)/\sqrt{1 + n/n_s}$$
$$\gamma(n) = \gamma_0 + \alpha(n) \cdot n \quad (2)$$

where $n_s$ is the saturation density of the coupling strength, $\alpha(n) = \alpha_0/(1 + n/n_d)$ represents the density-dependent broadening coefficient, with an initial value of $\alpha_0$ and saturation density of $n_d$.

All the polariton properties can be fitted well using the 2D exciton response model (Fig. 3d, e). The fitting $\alpha_0$ are $0.21 \pm 0.05$ and $0.36 \pm 0.11$ μeV μm$^2$ for X and XX$^-$, respectively, in good agreement with previous reports[23,49]. The comparable broadening of different couplings suggests a similar dephasing nonlinearity. The coupling strength of X remains almost constant below the density of $10^4$ μm$^{-2}$ and slightly decreases by ~20% at $10^5$ μm$^{-2}$, with $n_{sX} = (3.0 \pm 0.5) \times 10^5$ μm$^{-2}$. This value is comparable to that deduced from the exciton Bohr radius[51], with $1/a_{BX}^2$~$10^6$ μm$^{-2}$, confirming the high accuracy of our density calibration. In sharp contrast, the coupling strength of XX$^-$ exhibits a significantly stronger saturation at a much lower density down to $10^2$ μm$^{-2}$, and it drastically drops by ~40% when the density increases to $10^4$ μm$^{-2}$, indicating a low saturation density of $n_{sXX^-} = (5.3 \pm 0.8) \times 10^3$ μm$^{-2}$, ~56 times smaller than $n_{sX}$.

In view of the fact that the nonlinearity of the plasmon–exciton hybrid system is essentially derived from the exciton component[32], we further carry out similar but independent $R_c$ spectral measurements for pure monolayer WS$_2$, as discussed in Supplementary Note 5. Similar nonlinear behaviors for both the broadening and saturation properties

are found, where $\alpha_0$ are $0.07 \pm 0.02$ and $0.5 \pm 0.1$ μeV μm$^2$, and $n_s$ are $(8 \pm 1) \times 10^5$ μm$^{-2}$ and $(2.0 \pm 0.2) \times 10^4$ μm$^{-2}$ for X and XX$^-$, respectively. Again, the saturation density of XX$^-$ is ~40 times smaller than X, agreeing with the scale factor from the hybrid system.

Both the coupling polariton and pure exciton systems indicate a significantly large saturation nonlinearity of XX$^-$. To further intuitively describe this nonlinearity, we plot the nonlinear coefficient quantified as $g = d\Omega/dn$ in Fig. 3f, where $n$ is the individual density of different excitation species rather than the total one (see Supplementary Note 4 for details). With decreasing $n$, $g_X$ increases from $0.08$ μeV μm$^2$ and saturates to $0.14$ μeV μm$^2$ at $10^3$–$10^4$ μm$^{-2}$, in agreement with the reported values[18,23]. In comparison, $g_{XX^-}$ exhibits a much more drastic change, which increases from $0.8$ to $4$ μeV μm$^2$, with a saturation of $n < 10^2$ μm$^{-2}$. Thus, a nonlinearity enhancement of ~30 times is obtained for XX$^-$ compared to X.

We now have a phenomenological analysis of the exceptionally high nonlinearity of XX$^-$ polariton. Considering the charged nature of XX$^-$, there are mainly two mechanisms that contribute to the saturation nonlinearity: the phase space filling due to the Pauli exclusion principle, and the local depletion of the doping electron gas that is available to form charged complexes. Both mechanisms have been reported to explain the large nonlinearity of microcavity polaritons comprised of different excitonic complexes in layered TMDs[18–21,23–26]. Here, the local electron depletion effect might not be the dominant mechanism since another charged exciton complex, the trion, does not present a large nonlinear interaction ($0.2$ μeV μm$^2$, see Supplementary Fig. 7). Nevertheless, the absence of electron depletion in trion still cannot safely rule out its contribution to the nonlinearity of

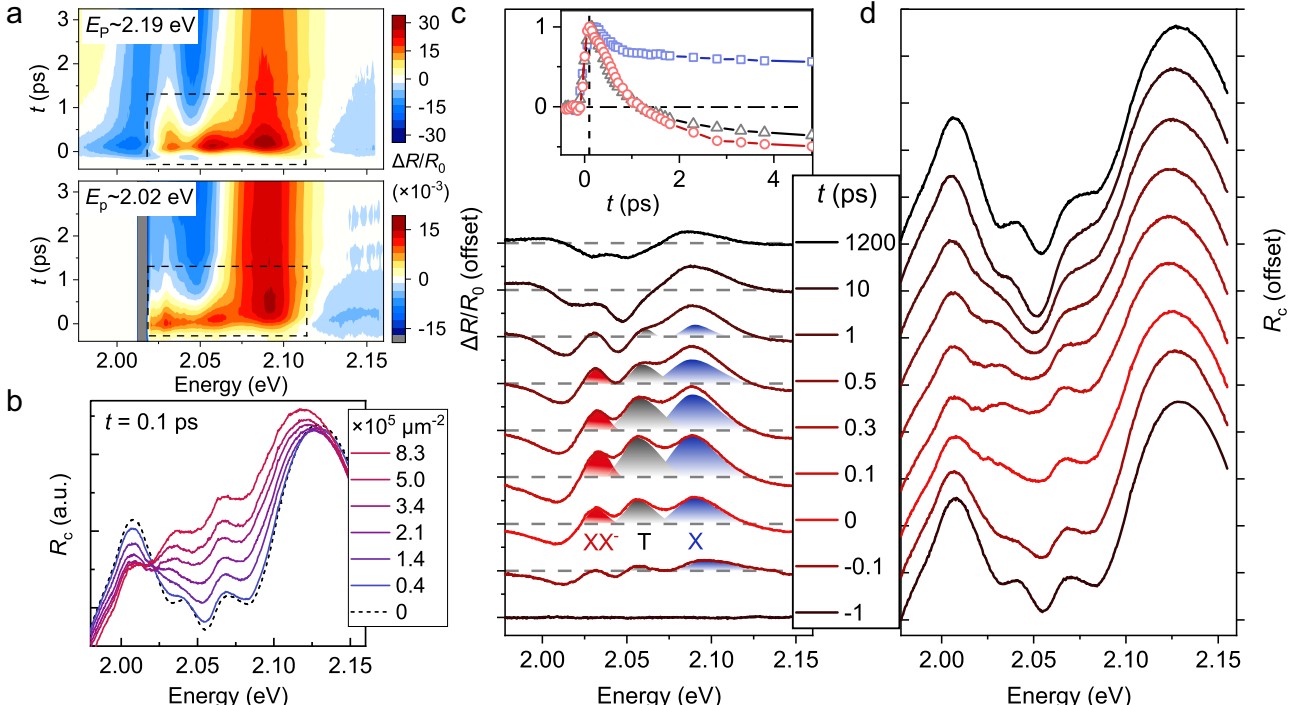

**Fig. 4 | Visualization of polariton nonlinearities and their dynamics. a** Transient absorption pseudo-color images of the coupling system with selected pump energy of ~2.19 eV (top panel) and ~2.02 eV (bottom panel). Zero-time delay is defined as the moment when the signal rises most rapidly. **b** Excitation density-dependent transient $R_c$ spectrum (solid lines) at $t = 100$ fs, deduced from $\Delta R/R_0$ spectrum in **a** and local steady-state $R_c$ spectrum without pumping (dash line). **c** and **d** $\Delta R/R_0$ (**c**) and transient $R_c$ (**d**) spectra at selected time delays. In **c**, the shaded absorption features ($\Delta R > 0$) represent the saturation strengths caused by corresponding pump injecting excited states, including both the coherent polaritons ($t < 100$ fs) and incoherent excitons ($100$ fs $< t < 1$ ps). While the X absorption feature at $t > 1$ ps is dominated by the lattice heating-induced band redshift, which manifests as a dispersive line shape. (Inset) The normalized dynamics of the three excitonic peaks. The vertical dash represents $t = 100$ fs.

XX⁻, as XX⁻ oscillator strength might be more sensitive to electron doping. Here, we preferably ascribe the strong saturation nonlinearity to the phase space filling in momentum space[15,19] (Fig. 3g), which provides a nonlinearity scaling to the square of the Bohr radius.

We further estimate the Bohr radius of XX⁻ ($a_{XX^-}$) based on the kinetic theory of thermalized exciton and the transient absorption dynamics of the three exciton complexes in monolayer WS₂ (Supplementary Note 6)[52]. Briefly, XX⁻ after the pump pulse duration is considered to be generated via the collision of X and T, with a formation rate $\beta_{XX^-}$ closely related to the Bohr radius: $\beta_{XX^-} = a_{XX^-}\sqrt{\nu_X^2 + \nu_T^2}$, where $\nu_{X(T)} = \sqrt{\frac{\pi k_B T}{2 m_{X(T)}}}$ is the 2D Maxwellian velocity of X(T) at 4 K, and $m_{X(T)} = 0.64(0.96)m_e$ is the X(T) mass. The $\beta_{XX^-}$ is estimated to be $0.8$ cm² s⁻¹ from fitting the exciton dynamics (Supplementary Fig. 16), corresponding to $a_{XX^-} \sim 5$ nm, which is comparable to the neutral biexciton cross-section of 4 nm in WSe₂[52]. Given that $n_s \propto 1/a^2$, a Bohr radius of 5 nm suggests a XX⁻ saturation density 25 times smaller than X, which is comparable, but smaller than the experimental values in the hybrid polariton (~56) and pure exciton (~40) systems. This implies a stronger XX⁻ nonlinearity than expected from the pure phase space-filling effect, possibly due to the modeling error or other factors, such as the additional contribution from the aforementioned doping depletion.

## Ultrafast dynamics of polaritons and excitons

While the above linear spectrum reveals the polariton nonlinearity through varying the incident intensity of the supercontinuum probe light, the nonlinear transient absorption (TA) spectrum can directly visualize such nonlinearity as well as its dynamics by introducing another monochromatic pump pulse and detecting the differential reflectance $\Delta R(t)/R_0 = (R(t) - R_0)/R_0$, where $R(t)$ and $R_0$ represent the probe reflectance with and without pumping, respectively, and $t$ is the

pump-probe delay (see the "Methods" section). Two representative examples are shown in Fig. 4a under pumping energies slightly larger than UP ($E_p \sim 2.19$ eV) and LP ($E_p \sim 2.02$ eV) resonances (see Supplementary Fig. 12 for more pumping cases). Within the first 1 ps when the polariton and exciton dominate the signal (discussed below), there is no notable difference between different pumping scenarios. Therefore, we focus on the $E_p \sim 2.19$ eV data below to prevent the influence of pump scattering.

Three distinct photo-induced absorption features (PA, $\Delta R > 0$) rise immediately after pumping, whose energies coincide well with the exciton resonances. To visually demonstrate the PA source, we deduce reversely the linear $R_c$ spectra from $\Delta R/R_0$ spectra at the peak delay of $t = 100$ fs at different injected densities, and compare them with the $R_c$ spectrum without pumping ($t = -1$ ps), as shown in Fig. 4b. Obvious saturation nonlinearity of the three exciton-polaritons is found, qualitatively resembling the nonlinear behavior in Fig. 3a. This suggests that the PA signal at the ultrafast timescale is a hallmark of the polariton nonlinearity, especially the saturation nonlinearity.

To further study the dynamics of the nonlinear behavior, Fig. 4c, d provide the measured $\Delta R/R_0$ and the deduced $R_c$ spectra at different time delays under a large pump density of $n_{tot} = 2.73 \times 10^5$ μm² (see Supplementary Fig. 17 for the case of small density). Remarkably, all the PA features increase immediately within the pump duration, and then peak at ~100 fs, when the coupling collapses most violently in the $R_c$ spectrum. After 100 fs, these PA peaks decay at different rates, manifesting as the recovery of the coupling strength. Specifically, the XX⁻ and T peaks completely disappear within a few ps and then further turn to photoinduced bleaching feature (PB, $\Delta R < 0$), while the X absorption peak lasts for a long time, until the measurement window of 1.2 ns. All these PA decay behaviors, including their density-dependent properties, are similar to the corresponding PB dynamics

in monolayer $WS_2$, where exciton-induced phase space filling and lattice heating-induced band redshift govern the spectral response at timescales of <1 and >1 ps[37,53], respectively (see discussions in Supplementary Note 6). Such similar dynamics imply that in the plasmon–exciton coupling system, the PA decay feature after 100 fs is predominantly caused by the incoherent exciton (100 fs–1 ps) and lattice heating (>1 ps), while only the ultrafast PA rise at the first ~100 fs is governed by the coherent polariton. This statement is further confirmed by the fact that the Rabi oscillation period is only $2\pi/\Omega$ ~ 40 fs for a typical $\Omega$ of 100 meV[41], much smaller than 100 fs. Nevertheless, considering that the hybrid system has a long coherent tail at the intermediate coupling regime[44], a delay of 100 fs can still be regarded as the longest time scale when the coherent polariton governs the TA signal. This critical timescale is well demonstrated by the dynamics of the UP resonance in the $R_c$ spectrum (Supplementary Fig. 18), where a maximum redshift caused by the saturation nonlinearity is found at 100–200 fs.

## Discussion

The low-lying energy state nature of $XX^-$ is practicable for real application compared with those highly energetic excited states such as 2 s Rydberg exciton[21]. The large binding energy of $XX^-$ (>50 meV) also ensures a considerable oscillator strength at an elevated temperature of >100 K, although the strong coupling condition is not satisfied here owing to the large cavity dissipation. Besides thermal stability, the most remarkable property of $XX^-$ polariton is its unimaginably high nonlinearity due to the intrinsically large Bohr radius. Note that a 30-fold enhancement of saturation coefficient should be a conservative value, since the doping depletion effect that generally contributes to the nonlinearity of charged excitonic complexes is relatively weak here, which might play a greater role in lightly doped cases[18]. However, it should be highlighted that $XX^-$ polariton in plasmonic nanocavity studied here is more dissipative than the one in dielectric microcavity in spite of the impressive nonlinearity, and thus it is not suitable to implement the regime of quantum optics (i.e., photon blockade). Instead, it possesses a sub-picosecond lifetime, which is of significant importance in the application of ultrafast modulation devices. Furthermore, our work tends to provide an introductory verification and principled explanation for polariton photophysics, where the conclusion is robust and gets reliable models, and we anticipate the so-called charged biexciton polariton is versatile in microcavity systems and capable of more sophisticated achievements. Moreover, the industrial quality and simple processing of the structure imply a high potential for mass, low-cost, scalable fabrication[54].

In summary, we have spectrally studied the plasmon–exciton polaritons in a $WS_2$–Ag ND hybrid system at different temperatures. The linear reflectance contrast spectrum, nonlinear transient absorption spectrum, as well as four-coupled oscillator model analysis provide comprehensive visualization of the high saturation nonlinearity of $XX^-$ polariton, which is 30 times higher than neutral exciton polariton. Large Bohr radius and probably the doping depletion effect might contribute to such high nonlinearity. In terms of universality, the $XX^-$ polaritons and their huge nonlinearity are also expected in microcavity structures such as distributed Bragg reflectors or photonic crystal resonators. In addition, considering the fermion nature of $XX^-$, electrical gate and external magnetic field[55,56] tunings may be direct and efficient strategies to control the oscillator strength as well as the many-body interaction of this kind of charged multiparticle polariton.

## Methods
### Sample preparation
Monolayer $WS_2$ synthesized by chemical vapor deposition (CVD) methods from Sixcarbon Technology was firstly transferred onto a precleaned fused silica substrate using the wet transfer technique. Then periodic Ag nanodisk (ND, 100 μm × 100 μm) arrays were directly etched and deposited on top of $WS_2$ monolayer by means of electron-beam lithography and evaporation. Finally, a 200 nm-thick polymethyl methacrylate (PMMA) layer was spin-coated onto the $WS_2$–Ag ND heterostructure to avoid sample degradation. The period of the Ag array is fixed at 300 nm, while the array diameter is varied from 80 to 140 nm to acquire different plasmon resonances.

### Optical measurements
**PL spectrum.** The sample is excited non-resonantly by a 532 nm CW laser, the reflection light including the PL and excitation components is collected by the microscope, and then the redundant excitation component is removed by a 532 nm long pass filter (Semrock). For spectral analysis, the pure PL is sent to a grating spectrograph (Shamrock 500i, Oxford Instrument) and then imaged by a plane array CCD camera (iVac 316, Oxford Instrument).

**Linear reflectance contrast ($R_c$) spectrum.** Both the linear $R_c$ spectrum and transient absorption spectrum below are carried out based on a Pharos femtosecond laser system (PH2–20W, Light Conversion, 1030 nm, 100 kHz, full-width at half-maximum of 230 fs and 20 W) and corresponding optical parameter amplifiers (ORPHEUS-HP and ORPHEUS-N–2H, Light Conversion). In the $R_c$ spectrum, only one supercontinuum white light pulse is used, which is generated by focusing a 700 nm, ~30 fs pulse (ORPHEUS-N–2H) onto a sapphire crystal. This white light is then normally focused onto the sample by a microscope, with a beam diameter of ~1.5 μm (width of 1/e intensity). The reflection beam is collected by the same microscope and separated from the incident light by a 90/10 beam splitter and finally is analyzed by the same spectrograph and camera used in the PL spectrum. The reflectance contrast spectrum is recorded as $R_c = (R_{sample} - R_{substrate})/R_{substrate}$, where $R_{sample}$ and $R_{substrate}$ are the reflectances of the sample and quartz substrate.

**Nonlinear transient absorption ($\Delta R/R_0$) spectrum.** $\Delta R/R_0$ spectrum is carried out on the basis of the $R_c$ spectrum, where another light pulse (ORPHEUS-HP) with narrow linewidth and large fluence (served as pump pulse) is focused onto the sample, and coincides exactly with the supercontinuum, small fluence pulse (served as probe pulse) used in the $R_c$ spectrum. In this scenario, the pump pulse pre-irradiates the sample and induces a mass of excited states instantaneously (plasmon, exciton, and their coupling system). The reflectance ($R$) of the subsequent probe pulse is modulated by these pump-induced excited states. Dynamics of the excited states are also acquired by changing the time delay of the two light pulses with a delay line, with a time resolution of ~200 fs, which is determined by the pulse width of the convolution of the two beams. The signal is recorded as $\Delta R(t)/R_0 = (R(t) - R_0)/R_0$, where $R(t)$ and $R_0$ are the probe reflectance with and without pumping. $\Delta R/R_0$ spectra under different pump energies are measured by tuning the photon energy and simultaneously removing the redundant pump component in the probe reflection by carefully choosing appropriate filter sets (Semrock), the setup sketch can be seen in Supplementary Fig. 19.

**Temperature-dependent experiment.** For low-temperature measurement, the sample is placed in a liquid-helium exchange gas cryostat (attoDRY1000, Attocube systems AG), equipped with a cryogenic compatible apochromatic objective (NA = 0.82) for confocal microscopy. For room temperature measurement, the sample is placed in air, with two collection objectives in reflection (NA = 0.55) and transmission (NA = 0.7) paths.

### Theoretical modeling
A four-coupled oscillator model is used to describe the reflection/absorption spectrum and anticrossing behavior of the four polariton eigenstates in the $WS_2$–Ag hybrid sample. By solving the rate equation

(see Supplementary Note 3 for details), the reflectance of the plasmon–exciton coupling system can be written as

$$R_1(\omega) \propto \mathrm{Im}(F_0 \dot{x}_{\mathrm{pla}}) \propto \omega\, \mathrm{Im}\left\{ \frac{S_X S_T S_{XX^-}}{S_{\mathrm{pla}} S_X S_T S_{XX^-} - \omega^2 \Omega_X^2 S_T S_{XX^-} - \omega^2 \Omega_T^2 S_{XX^-} S_X - \omega^2 \Omega_{XX^-}^2 S_X S_T} \right\} \tag{3}$$

where $S_j = \omega_j^2 - \omega^2 - i\gamma_j\omega$ ($j =$ pla,X,T or XX$^-$).

Remember that Eq. (3) can only describe a coupled system in which Ag plasmons are in close contact with WS$_2$ excitons. While in our sample, the plasmons are mainly excited near the up and below edge rings of Ag ND, and only plasmons near the below ring can couple with the excitonic complex (the thickness of the ND is ~30 nm, much larger than plasmon–exciton coupling length). Thus, a Lorentz shape base should be considered in fitting the reflection spectrum, accounting for the spatially separated plasmons near the up ring, with

$$R_2(\omega) \propto \omega\, \mathrm{Im}\left\{ \frac{1}{S_{\mathrm{pla}}} \right\} \tag{4}$$

The reflection spectrum of the whole system can thus be expressed as

$$R(\omega) = R_{\mathrm{pol}}(\omega) + R_{\mathrm{pla}}(\omega) \propto k R_1(\omega) + (1-k) R_2(\omega) \tag{5}$$

where $R_{\mathrm{pla}}(\omega)$ and $R_{\mathrm{pol}}(\omega)$ describe the pure uncoupled plasmon and plasmon–exciton coupling components, respectively, and $k \in 0$–$1$ represents the coupling proportion.

The anticrossing behaviors and Hopfield coefficients of the four polariton eigenstates are obtained by numerical solving the determinant of the coefficient matrix of the rate equation, which is equivalent to diagonalizing the Hamiltonian shown in Eq. (1).

## Data availability
The datasets generated during and/or analyzed during the current study are available from the corresponding author on request. Source data are provided with this paper.

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

## Acknowledgements
The authors are grateful for financial support from the National Natural Science Foundation of China (62105364, 62075240); The Science and Technology Innovation Program of Hunan Province (2021RC2068); The Scientific Researches Foundation of the National University of Defense Technology (ZK22-16); and Postgraduate Scientific Research Innovation Project of Hunan Province (CX20220001).

## Author contributions
T.J. and K.W. conceived the project. Q.L. and K.W. established the spectral setup and performed the measurements, with help from Y.Y., Y.T. fabricated the samples. K.W., Q.L., Y.T., Z.X., and T.J. analyzed the data and interpreted the results. Z.X., K.W., and Q.L. built the model and performed the simulations. Q.L., K.W., Y.T., and T.J. wrote the manuscript with input from all the authors. All the authors discussed the results.

## Competing interests
The authors declare no competing interests.
