## [Peer Review File · Nature Communications]

Reviewers' Comments:

Reviewer #1:

Remarks to the Author:

The authors' study examines the nonlinearity of the plasmon-exciton-trion-charged biexciton coupling state in silver nanocavities and presents the transient absorption dynamics of biexciton polaritons, with a timescale of less than 100 fs. This study presents interesting findings, but a major technical issue needs to be addressed before it can be published.

Firstly, the authors refer to the resultant quasiparticles as polaritons, even though the coupling strength is not strong enough to warrant such a classification, especially for the charged biexciton polaritons that are the main focus of the study.

Secondly, the authors suggest in the discussion part that similar physics should also be observed in a dielectric planar microcavity with at least one order of magnitude lower linewidth, which would allow for strong light-matter coupling compared to the intermediate coupling achieved in this study. It is unclear why the authors chose a plasmon cavity with a lower Q cavity mode over a standard higher Q planar microcavity.

Thirdly, the authors claim in the abstract that the polaritons inherit the coherence of light, which is not correct. It is unclear what the authors mean by this statement. Polaritons emit coherent light only if they form a condensate.

Finally, the authors' comment that coherent polaritons last for the first 100 fs and incoherent excitons exist for $100 \text{ fs} < t < 1 \text{ ps}$ requires further clarification. It is not clear what the difference is between coherent polaritons and incoherent excitons in the experiment.

Overall, while the study presents exciting results, these technical issues need to be addressed before it can be recommended for publication in Nature Communications.

Reviewer #2:

Remarks to the Author:

The work reports on the nonlinear properties of transition metal dichalcogenides (TMDs), specifically WS₂ monolayers, coupled to silver nanodisks (Ag NDs). The reflectance contrast (R_c) of the WS₂ – Ag ND is measured as a function of different cavity sizes, showing coupling between different excitonic complexes and the plasmonic resonances. Performing power- and temperature-dependent studies, the authors focus on the nonlinearity associated with charged biexcitons (XX⁻) and conclude that it is ~ 30 times higher than that of neutral excitons (X), mainly because of the phase space filling effect. Finally, by means of transient absorption spectra, the authors also dig the dynamics of the coupled system.

The text is generally well referenced, often citing previous literature in order to support statements and confirm achieved results. From this point of view, the introduction provides a general overview of the light-matter coupling field, allowing the reader to place this work in a proper context.

The present work shows an interesting approach to exploit TMDs nonlinear characteristics, especially looking at XX⁻, which have been little studied so far. The topic can be further deepened using other photonic structures and experimental techniques for more practical applications (as suggested by the authors), but this work paves the way towards this direction.

Data are most of the time consistent and largely well-presented. However, there are a few questions that should be addressed:

1. Looking at Supplementary Figure 1 (bottom-left panel, for the heterostructure with array diameter of 90 nm), there is an energy shift of the three photo-induced absorption features representing the excitonic resonances, as also emphasized by the three dashed vertical lines, compared to the other two bottom panels for bigger NDs. Since the presence of the cavity should not affect the bare excitonic energies, the shift could be due to some inhomogeneity in the WS₂

monolayer. But this should then have an impact on the definition of the detuning (plasmon energy - exciton energy, i.e., $E_{pl} - EX$), due to the fact that measurements on NDs with different sizes are needed to tune E_{pl} . NDs that are deposited on top of inhomogeneous WS₂ monolayer, with consequent variation of EX at the same time. Nevertheless, this effect is not clearly manifested in Figure 1c and Figure 1d of the main text. More details should be included in the manuscript about this.

2. According to the authors, the definition of the differential reflectance ($\Delta R(t) / R_0 = (R(t) - R_0) / R_0$, where $R(t)$ and R_0 represent the probe reflectance with and without pumping, respectively, and t is the pump-probe delay') provided at line 213 of the main text ("Ultrafast dynamics of polaritons and excitons" section) implies that $\Delta R > 0$ is photo-induced absorption and $\Delta R < 0$ is bleaching. From my understanding, they should actually be the other way round, i.e., bleaching when $\Delta R > 0$ and absorption when $\Delta R < 0$. Nevertheless, I would like to stress that this observation does not affect the validity of the transient absorption measurements presented in the current manuscript.

3. Greater perplexity about transient absorption data comes from Supplementary Figure 16a, where the authors observe opposite behavior for the TMDs when off-cavity and in-cavity. Specifically, they detect what they define as absorption ($\Delta R > 0$ in the current version of the manuscript) for the three excitonic resonances for the in-cavity configuration (Figure 4a in the main text), while bleaching ($\Delta R < 0$) for the bare monolayer, i.e., off-cavity configuration. According to the results reported in "Adv. Optical Mater. 2020, 8, 2001147", there should be no switching from absorption to bleaching for TMDs when inside or outside a cavity. More information are required to explain the opposite phenomenon for the off- and in-cavity configuration.

4. In the rate equation S1 representing the whole coupled system, from my understanding from "American Journal of Physics 78, 1199-1202 (2010)", all the terms presenting the different coupling strengths Ω should be multiplied by the displacement x , not its derivative.

5. Finally, in Supplementary Figure 18, it is not clear what the step size of the time delay setup is, since the dense set of data points shown, but a time resolution of ~ 100 fs is declared in the Method section (3. Nonlinear transient absorption ($\Delta R/R_0$) spectrum). From that point of view, some more details about the experimental setup, accompanied by a sketch, would be really useful to guide the reader through the manuscript.

Overall, the main conclusions are well supported by the above mentioned comprehensive studies, where the data are reinforced by valid theoretical models.

Some minor improvements might be necessary to facilitate readers' understanding:

- In Figure 1c of the main text, label at least top and bottom curves, referring to E_{pl} of Figure 1d, as done in Figure 2b with temperature.
- In Figure 3a of the main text, label at least top and bottom curves, referring to the total polariton density n_{tot} .
- In Figure 3g of the main text, the circle in the trap states should be red, since electrons populate those levels. Of course, same thing applies for the corresponding Supplementary Figure 4a, where the three sub-figures should be labelled.
- Lines 76 – 78 in Supplementary Information related to the energy of the absorbed photons being resonant and then smaller than that of XX^- . The sentences here are a little bit misleading, so it would be better to rephrase and clarify the XX^- formation process.
- In Supplementary Figure 10 (left panels, for the heterostructure), it should be better to clearly explain the meaning of orange, red and blue curves in the corresponding caption.
- In Supplementary Figure 16 caption, according to the legend of panel (c), the orange curve should be with the band redshift, while the blue curve should be without considering the band redshift. Legend and caption are then not consistent with each other.

Reviewer #3:

Remarks to the Author:

In the manuscript, the authors present a spectroscopic study of a monolayer WS₂ crystal coupled to a silver nanocavity. They show the emergence of various resonances in reflectance contrast measurements, which they attribute to exciton- (X), trion- (T) and charged biexciton- (XX^-) polaritons. They present measurements of the nonlinearity and claim an "unimaginably high

nonlinearity" arising from the XX- resonances, promising "more quantum achievement" in the future.

The topic of the article is certainly timely, as it address the important question how the nonlinearity of otherwise weakly interacting excitonic systems can be increased. Coupling to a silver particle is also an interesting idea as that could potentially translate excitonic nonlinearity into photon interactions.

However, I have several major concerns:

1. From the outset, it is not clear why the authors target the polariton response rather than the pure excitonic nonlinearity. Is the silver "cavity" only to enhance the coupling strength?
2. The authors make strong claims about applications, in particular quantum applications of their work. However, no quantum effects are demonstrated and the exact prospects are not made clear. I think the authors should moderate those claims and instead clearly state the results that they do have.
3. Throughout the article, I find the argumentation not easy to follow. In particular, the authors give one possible explanation for their observations without backing up against literature or thoroughly evaluating alternatives. This begins with the assignment of the linear peaks as X, T and XX-, continues with ad-hoc assumptions about the origins of the nonlinearity and finishes with the rate model in (S11).
4. To determine nonlinearities, the polariton densities have to be determined. Why does the polariton lifetime not appear in S6?
5. The presented theory often relies on crude estimates, which may be sufficient for a qualitative understanding but (probably) do not support some of the conclusions the authors draw. For example: "Given that $n_s \propto 1/a^2$, a blockade radius of 5 nm suggests a XX- saturation density 25 times smaller than X, which is comparable, but smaller than the experimental values in the hybrid polariton (~ 56) and pure exciton (~ 40) systems. This implies a stronger XX nonlinearity than expected from pure polariton blockade, possibly due to the additional contribution from the aforementioned doping depletion." This discrepancy may give some indication but the estimate " $n_s \propto 1/a^2$ " simply does not allow finer conclusions without a more detailed analysis.

Minor comments:

- 1) ND is not defined in the text, and neither is PB.
- 2) Regarding the identification of peaks? Have the authors performed measurements to determine the charge state of the resonances?
- 3) What are the error bars in $1d/e$?
- 4) There are no error bars in 2b. Also, the theory curves fit perfectly, because new parameters (energy, γ , Ω) are used at every T and fluence. Given so many fit parameters, the predictive value seems limited. Can their values be fixed globally (possibly within a temperature-dependent model)?
- 5) "Note that Ω_X at 293 K is in excellent agreement with $\sqrt{\Omega_X^2 + \Omega_T^2 + \Omega_{XX}^2} = 138$ meV at 4 K [...]" What does this mean?

Comments from Reviewer #1

Reviewer #1 (Remarks to the Author):

The authors' study examines the nonlinearity of the plasmon-exciton-trion-charged biexciton coupling state in silver nanocavities and presents the transient absorption dynamics of biexciton polaritons, with a timescale of less than 100 fs. This study presents interesting findings, but a major technical issue needs to be addressed before it can be published.

Overall, while the study presents exciting results, these technical issues need to be addressed before it can be recommended for publication in Nature Communications.

Comment 1. Firstly, the authors refer to the resultant quasiparticles as polaritons, even though the coupling strength is not strong enough to warrant such a classification, especially for the charged biexciton polaritons that are the main focus of the study.

Response1: We appreciate Reviewer #1 for his/her careful consideration and meaningful question. The raised question here is very profound and valuable at the heart of the research field, especially for an exciton-polariton system based on plasmonic nanocavity, also referred as plexciton in the majority of literature. As we claimed in the main text, we confirmed the formation of a polaritonic state in our 2D semiconductor-nanocavity heterostructure according to the numerical relationship of coupling strength (i.e., Ω_{X,T,XX^-} and algebraic difference/sum of plasmonic and excitonic linewidths (i.e., $(\gamma_{\text{pla}}-\gamma_{X,T,XX^-})/2$ or $(\gamma_{\text{pla}}+\gamma_{X,T,XX^-})/2$). According to the reviewer's comments, we realized that there was no uniform evaluation of the coupling criteria in previous manuscripts, which may lead to confusion in definition. Therefore, we conducted a careful investigation and revision based on the literature (*Opt. Exp.* 2010, 18, 23633). Specifically, a broadly-defined plasmon-exciton polariton has a coupling strength larger than $(\gamma_{\text{pla}}-\gamma_{X,T,XX^-})/2$. While if the coupling strength is slightly smaller than $(\gamma_{\text{pla}}-\gamma_{X,T,XX^-})/2$ but much larger than the excitonic dissipation rate, the coupling state is called intermediate coupling. In this coupling state, the spectral splitting is already apparent in frequency domain, but the Rabi oscillation in the time domain is still obscured by the large damping of the plasmonic mode. Such very interesting and broadly defined moderate polaritonic regime has been extensively reported in recent years, especially for plasmon/exciton hybrid systems (*ACS Nano* 2018, 12, 10393; *Phys. Rev. B* 2020, 102, 205409).

In detail, we found that $\Omega_{X,T}$ (88, 80 meV) is slightly bigger than $(\gamma_{\text{pla}}-\gamma_{X,T})/2$ (77, 77.5 meV) and far beyond $\gamma_{X,T}$ (26, 25 meV), which suggests that a dominating mode splitting ($\Omega > (\gamma_{\text{pla}}-\gamma_{X,T})/2$) effect should contribute to the strong coupling between the plasmon and exciton/trion (X, T) here. On the other hand, for the state of charged biexciton (XX^-), the coupling strength Ω_{XX^-} (70 meV) is smaller than but close to $(\gamma_{\text{pla}}-\gamma_{XX^-})/2$ (81 meV), indicates that plasmon-exciton coupling states formed in

our system are in the intermediate coupling regime (i.e., on the border of Fano interference and mode splitting). At this moment, spectral splitting is already apparent in the frequency domain, while the Rabi oscillations in the time domain are still obscured by the large damping of the plasmonic mode. Even though the oscillating feature will disappear quickly, such an intermediate coupling state is reported to show a long-lasting tail in the time domain that is still coherent (*Nanophotonics*, 2020, 9, 3587). Thus, the formed XX^- hybridized state in Ag nanocavity of this work can still be recognized as a generalized polariton state.

Moreover, it has been detailedly discussed in *Light: Sci. Appl.* 2022, 11, 94 (refer to Supplementary Figure S19) that the coupling strength of the whole system will not profoundly influence the nonlinearity under certain conditions. In other words, the plasmon-exciton states in both the intermediate and the very beginning of strong coupling regime have the same nonlinear response mechanism. In our work, we focus on polaritonic nonlinearity rather than the view of coupling strength, where intermediate coupling and strong coupling are both involved and show no significant difference in nonlinear behaviors (Supplementary Figure S1). Altogether, the polaritonic study for the states of excitonic complexes, with the consistent parameters as well as the nonlinear characterization is in a certain degree of rationality.

Revision 1: We have revised the relevant description, from “As a result, only intermediate coupling regime is reached here for X and T, with $(\gamma_{\text{pla}} - \gamma_{\text{X,T}})/2 < \Omega_{\text{X,T}} < (\gamma_{\text{pla}} + \gamma_{\text{X,T}})/2$. While for XX^- , the Ω_{XX^-} is slightly smaller, but comparable to $(\gamma_{\text{pla}} - \gamma_{XX^-})/2$, and far beyond γ_{XX^-} , suggesting a hybridization of Fano interference and mode splitting, which can also be characterized under the polariton framework.” to “As a result, only strong coupling regime is reached here for X and T, with $\Omega_{\text{X,T}} > (\gamma_{\text{pla}} - \gamma_{\text{X,T}})/2$. While for XX^- , the Ω_{XX^-} is slightly smaller, but comparable to $(\gamma_{\text{pla}} - \gamma_{XX^-})/2$, and far beyond γ_{XX^-} , suggesting a hybridization that is in coherence at initial excitation composed of mode splitting ($\Omega_{\text{X,T}} > (\gamma_{\text{pla}} - \gamma_{\text{X,T}})/2$) and Fano interference ($\Omega_{\text{X,T}} < (\gamma_{\text{pla}} - \gamma_{\text{X,T}})/2$). This coupling strength, also known as intermediate coupling, has been extensively studied in recent years and can also be characterized by the polariton framework.” with the added Reference 43.

Comment 2. Secondly, the authors suggest in the discussion part that similar physics should also be observed in a dielectric planar microcavity with at least one order of magnitude lower linewidth, which would allow for strong light-matter coupling compared to the intermediate coupling achieved in this study. It is unclear why the authors chose a plasmon cavity with a lower Q cavity mode over

a standard higher Q planar microcavity.

Response 2: The planar microcavity with a high Q factor and low loss (narrow linewidth) is certainly a more popular platform in academia and suitable for quantum physics. By a distinct route, in our Ag-WS₂ system, the Ag nano cavity has a mode volume beyond the optical diffraction limit to compress the scale range of light-matter interaction, which undoubtedly promotes strong coupling states close to polaritons (*Nat. Commun.* 2018, 9, 801; *ACS Nano* 2022, 1612711), especially considering that the research subject in the manuscript, XX⁻ has a larger Bohr radius than X and T in theoretical terms. Although the cavity photons of plasmon nanocavities have a large loss, the accompanied large linewidth also facilitates to observe coupling states of three excitonic complexes simultaneously in a single spectrum, which may not be achieved by planar microcavities that require angle resolution and resonantly tuning of photon energies. At the same time, plasmon-exciton polaritons in Ag-WS₂ hybrid system exhibit ultrafast optical response, making them highly suitable for the preparation of nonlinear optics based on ultrafast laser correlation (*Phy. Rev. Lett.* 2021, 126, 117402; *Light: Sci. Appl.* 2022, 11, 94).

Comment 3. Thirdly, the authors claim in the abstract that the polaritons inherit the coherence of light, which is not correct. It is unclear what the authors mean by this statement. Polaritons emit coherent light only if they form a condensate.

Response 3: The quasiparticle polariton we described is in a coherent state, with the process of light-matter interaction does not involve the subsequent emitting processes. Our study has not concerned about the plasmon influence on exciton emission in the Ag-WS₂ system, different from some previous works (e.g. *ACS Nano* 14, 13841–13851 (2020)). The coherent state means that the exciton and photon phases are in coherence within the time scale of the existence of polaritons. In the transient absorption spectrum of the main text Figure 4, it can be clearly seen that the time scale of polariton response is $\sim 10^2$ fs, much faster than $10\sim 10^2$ ps of exciton luminescence (*2D Mater.* 2022, 9, 015023) according to the conclusion of time-correlated single photon counting (TCSPC). Therefore, the polaritonic coherent state we mentioned refers to the phase coherence characteristics in the excitonic formation process, instead of polariton condensate and luminescence inherited the coherence from light.

Comment 4. Finally, the authors' comment that coherent polaritons last for the first 100 fs and incoherent excitons exist for $100 \text{ fs} < t < 1 \text{ ps}$ requires further clarification. It is not clear what the difference is between coherent polaritons and incoherent excitons in the experiment.

Response 4: In optical measurement, it is crucial to elucidate the ultrafast response of the light-matter interaction system, especially for a coherent exciton-plasmon coupling state with so fast

lifetime. As reported in *Nat. Photonics* 2013, 7, 128, the polaritonic damping (dephasing) time in an exciton-plasmon coupling system should be <100 fs (shorter than laser pulse width and temporal resolution), thus a typical ~ 100 fs polariton lifetime is identified in the manuscript. The short lifetime can merely support one cycle of coherent energy transfer between exciton and plasmon, thus a short Rabi oscillation. After this, plasmon still tends to continue the transfer process, while the relaxation rate ($\sim 10^1$ fs corresponding to linewidth over 100 meV) is much faster than that of excitons, thus the phase of exciton and plasmon are out of synchronization and the coherent energy transfer is cut off. For a longer timescale, the excitation energy is mainly stored in an exciton reservoir with the presence of incoherent photon exchange between exciton and plasmon (*ACS Nano* 2014, 8, 1056), then the system presents the combined response from incoherent plasmon and exciton. The dynamics can be seen in Fig. R1 (reproduced from *Light: Sci. & Appl.* 2022, 11, 94)

Figure R1. Schematic of exciton-plasmon polariton (plexciton) dynamics after resonant pump pulse excitation. Figure reproduced from *Light: Sci. & Appl.* 2022, 11, 94.

Comments from Reviewer #2

Reviewer #2 (Remarks to the Author):

The work reports on the nonlinear properties of transition metal dichalcogenides (TMDs), specifically WS₂ monolayers, coupled to silver nanodisks (Ag NDs). The reflectance contrast (R_c) of the WS₂ – Ag ND is measured as function of different cavity sizes, showing coupling between different excitonic complexes and the plasmonic resonances. Performing power- and temperature-dependent studies, the authors focus on the nonlinearity associated with charged biexcitons (XX-) and conclude that it is ~ 30 times higher than that of neutral excitons (X), mainly because of phase space filling effect. Finally, by means of transient absorption spectra, the authors also dig the dynamics of the coupled system.

The text is generally well referenced, often citing previous literature in order to support statements and confirm achieved results. From this point of view, the introduction provides a general overview of the light-matter coupling field, allowing the reader to place this work in a proper context.

The present work shows an interesting approach to exploit TMDs nonlinear characteristics, especially looking at XX-, which have been little studied so far. The topic can be further deepened using other photonic structures and experimental techniques for more practical applications (as suggested by the authors), but this work paves the way towards this direction.

Data are most of the time consistent and largely well-presented. However, there are a few questions that should be addressed:

Comment 1. Looking at Supplementary Figure 1 (bottom-left panel, for the heterostructure with array diameter of 90 nm), there is an energy shift of the three photo-induced absorption features representing the excitonic resonances, as also emphasized by the three dashed vertical lines, compared to the other two bottom panels for bigger NDs. Since the presence of the cavity should not affect the bare excitonic energies, the shift could be due to some inhomogeneity in the WS₂ monolayer. But this should then have an impact on the definition of the detuning (plasmon energy - exciton energy, i.e., $E_{\text{pla}} - E_{\text{X}}$), due to the fact that measurements on NDs with different sizes are needed to tune E_{pla} . NDs that are deposited on top of inhomogeneous WS₂ monolayer, with consequent variation of E_{X} at the same time. Nevertheless, this effect is not clearly manifested in Figure 1c and Figure 1d of the main text. More details should be included in the manuscript about this.

Response 1: Thanks very much for the reviewer's suggestion. When calculating the data in Fig. 1c and Fig. 1d, the y-axis represents the energy of each polaritonic branch and is directly extracted from the experimental data, while the x-axis represents the energy of the plasmon in the nanocavity, which is calculated based on the following relationship:

$$E_{LP} + E_{MP1} + E_{MP2} + E_{UP} = E_{pla} + E_X + E_T + E_{XX^-} \quad (R1)$$

the left side of the equation represents the energy of four polariton branches, while the right side of the equation represents the sum of plasmon energy (E_{pla}) and three types of excitonic energy (E_X , E_T , E_{XX^-}). Strictly speaking, as the reviewer stated, due to the sample inhomogeneity, E_X , E_T and E_{XX^-} vary with the measurement position (i.e., the different points in Fig. 1c and Fig. 1d), so this inhomogeneity needs to be taken into account in the E_{pla} calculation. However, in fact, we are unable to measure the in situ excitonic energy in the heterostructure (i.e. the different points in Fig. 1c and Fig. 1d). Therefore, the commonly used method in literature (*Nano Lett.* 2018, 18, 1777-1785) is to determine E_X from the controlled pure WS₂ region at the same batch of materials, while ignoring the sample inhomogeneity, which is also the method we adopted here.

To further confirm the rationality of this approximation, we conducted R_c measurements on different regions of pure WS₂ from the same batch (Fig. R2a), and solved the spectrum with three Lorentz peaks representing X, T, and XX⁻ resonances (Fig. R2b). It can be seen that due to the inhomogeneity of the sample, although the shape of R_c changes (mainly caused by changes in the intensity ratio of the three complexes), the excitonic energy ($\Delta E_{X,T,XX^-}$) shift very little, within 5meV, which is much smaller than the dispersion of the polaritons, even for those polaritons with subtle changes, such as MP1 ($\Delta E_{MP1}=E_T-E_{XX^-}=24$ meV) and MP2 ($\Delta E_{MP2}=E_X-E_T=29$ meV). As a result, the sample inhomogeneity can be safely ignored when determining E_{pla} using Equ. R1. To illustrate this visually, based on Fig.1c, we also plot the excitonic energies from different WS₂ regions, and compared them with the polariton dispersions, as shown in Fig. R2c. It can be clearly found that the influence of sample inhomogeneity (dots) on the anticrossing dispersion (dashes) can be slight.

Finally, it should be noted that the dip position (dashed line) of the transient absorption spectrum in the Ag-WS₂ region in Fig. 1 (as shown in the Supplementary Fig. 1) is not exactly equivalent to the exciton resonance of R_c , which is significant in off-resonance case. In theory, as shown in Fig. R2d, the actual excitonic resonant energy (E_X) may lie somewhere between the maximum and the minimum of the Fano profile (dispersion curve), whose relative deviation can be defined by the asymmetry parameter q (*Rev. Mod. Phys.* 2010, 82, 2257). It can be seen that in asymmetric situation of $q=1$, the spectral dip gets an evident deviation from resonant energy of $\varepsilon=0$, where $\varepsilon = 2(E - E_X) / \gamma_X$, and γ_X is the excitonic linewidth.

Fig. R2 Homogeneity of WS₂ samples. **a** R_c spectra from nine regions of pure WS₂ from the same batch. **b** X, T and XX⁻ resonances versus different points extracted from **a**. **c** The demonstration of resonance fluctuation from **b** plotted to compare with the dispersion curve. The non-uniformity of this sample can be ignored relative to the constant resonant value (horizontal dashed line). **d** Normalized Fano profiles for various values of the asymmetry parameter q , figure reproduced from *Rev. Mod. Phys.* 2010, 82, 2257.

Revision 1: We have supplemented the description, “Here, the shift of the excitonic resonance caused by sample inhomogeneity is neglected since it is much smaller than the energy change scale of polaritons in the dispersion curves (Supplementary Fig. 20).”, with the figure adding into Supplementary Information.

Supplementary Figure 20. Homogeneity of the sample. (a) R_c spectra of nine different WS₂ regions from the same batch of materials. (b) Extracted resonant energy of X, T and XX⁻ from (a) by Lorentzian fitting. (c) Comparison between the polariton dispersions from Fig. 1d and the plotted excitonic energies from (b).

Comment 2. According to the authors, the definition of the differential reflectance ($\Delta R(t) / R_0 = (R(t) - R_0) / R_0$, where $R(t)$ and R_0 represent the probe reflectance with and without pumping, respectively, and t is the pump-probe delay') provided at line 213 of the main text ("Ultrafast dynamics of polaritons and excitons" section) implies that $\Delta R > 0$ is photo-induced absorption and $\Delta R < 0$ is bleaching. From my understanding, they should actually be the other way round, i.e., bleaching when $\Delta R > 0$ and absorption when $\Delta R < 0$. Nevertheless, I would like to stress that this observation does not affect the validity of the transient absorption measurements presented in the current manuscript.

Response 2: We thank the reviewers for their questions. In fact, differential reflectance ' $\Delta R(t)/R_0=(R(t)-R_0)/R_0$ ' is derived from the change of the sample reflectivity $R(t)$ caused by the disturbance of pump light. Therefore, this is directly related to the reflectivity $R(t)$ of the sample compared with the substrate. The substrates often used are transparent fused silica substrates and opaque Si/SiO₂ substrates. Assuming that the sample absorption is α , reflectivity is R , and transmittance is T , then:

In transparent fused quartz substrate, if α is enhanced, R increases, while T decreases to ensure $\alpha+R+T=1$ (ignoring the absorption of the substrate). Therefore, α is in phase with R , with $\Delta R/R_0 > 0$ representing photoinduced absorption (PA) peak.

In comparison, in the opaque Si/SiO₂ substrate, due to $T = 0$ and eliminating the substrate's absorption, $\alpha+R \approx 1$. Therefore, if α increases, i.e. a PA peak, R may decrease, with $\Delta R/R_0 < 0$.

The substrate used in this article is a transparent fused quartz substrate, thus $\Delta R/R_0 > 0$ indicates a photo-induced absorption feature, while $\Delta R/R_0 < 0$ represents a bleaching one. This conclusion is also consistent with the literature (*Nat. Photon.* 2015, 9, 466), where WS₂ is placed on a transparent fused quartz substrate, as shown in Fig. R3.

Fig. R3 Photoinduced optical response of WS₂. **a** R_c , transmission and resulting absorption spectra of monolayer WS₂ at room temperature. **b** The comparison of R_c with/without pump excitation. Figure reproduced from *Nat. Photon.* 2015, 9, 466.

Comment 3. Greater perplexity about transient absorption data comes from Supplementary Figure 16a, where the authors observe opposite behavior for the TMDs when off-cavity and in-cavity. Specifically, they detect what they define as absorption ($\Delta R > 0$ in the current version of the manuscript) for the three excitonic resonances for the in-cavity configuration (Figure 4a in the main text), while bleaching ($\Delta R < 0$) for the bare monolayer, i.e., off-cavity configuration. According to the results reported in "Adv. Optical Mater. 2020, 8, 2001147", there should be no switching from absorption to bleaching for TMDs when inside or outside a cavity. More information is required to explain the opposite phenomenon for the off- and in-cavity configuration.

Response 3: Thanks for the reviewer's question. Here, the transient absorption spectrum is carried out in reflective geometry, with $\Delta R(t)/R_0 = (R(t) - R_0)/R_0$, where R_0 is the reflectivity of the sample with (without) pump. According to **Response 2**, in the controlled WS₂ region, when there is no pump, the R_0 is an upper convex Lorentzian type larger than zero, as shown in Fig. R4a (blue region, only one single peak is illustrated here for simplicity). After pumping, if we only consider the state-filling induced saturation nonlinearity, the absorption (R) is weakened, as shown in Fig. R4a (red region). Therefore, a photoinduced bleaching (PB) signal is acquired, with $\Delta R/R_0 < 0$, as shown in Fig. R4a (lower panel, orange region).

By contrast, the signal in the Ag-WS₂ heterostructure is completely different. When there is no

pump (for simplicity, considering the case of complete resonance between the cavity and exciton), R appears as an upward peak with a Fano-type dip, as shown in Fig. R4a (blue area), in which the dip amplitude represents the coupling strength between the cavity and exciton. After pumping, the coupling strength weakens and the spectra dip becomes smaller due to saturation nonlinearity, as shown in the red area of Fig. R4b. In this way, the obtained saturation signal is a PA peak, with $\Delta R/R_0 > 0$ (orange area in the lower panel of Fig. R4b), opposite to the pure excitons in monolayer WS_2 .

Fig. R4 Origin of opposite $\Delta R/R_0$ between excitonic and polaritonic state. **a** Upper: Schematic of the saturation effect that weakens the exciton oscillator strength and decreases the reflection R . Lower: negative PB feature is acquired through $\Delta R/R_0$ calculation. **b** Upper: Schematic of the saturation effect that weakens the polariton coupling strength (spectral dip) and increases the reflection R . Lower: a positive PA peak of $\Delta R/R_0$.

The experimental spectra are shown in Fig. R5, providing mutual corroboration for the schematic paradigm above. However, due to the fact that the actual signal not only includes the state filling-induced saturation effect but also includes linewidth broadening and energy shift, etc., there will be PA signals appearing at both sides of the excitonic resonance in monolayer WS_2 , as well as PB signals in Ag- WS_2 hybrid system.

Fig. R5 Measured R_c and $\Delta R/R_0$ in experiment. Compared with Fig. R4, the saturation effect exhibits similar response in R_c and $\Delta R/R_0$ for both WS_2 (exciton) and $Ag-WS_2$ (polariton) samples, while X, T, XX^- coexist in the low-temperature condition (4 K).

In summary, the state-filling induced saturation in pure WS_2 and strong coupling $Ag-WS_2$ heterostructure should appear to be PB ($\Delta R/R_0 < 0$) and PA ($\Delta R/R_0 > 0$) signals in transient absorption spectrum, respectively. This point has also been proven in previous literature, such as the work from Qihua Xiong's group (*ACS Photon.* 2019, 6, 2832), as shown in Fig. R6.

Fig. R6 Ultrafast dynamics of the exciton–plasmon coupling in the WS₂–Ag hybrid structure in comparison to the uncoupled subsystems. a–c transient absorption signals for the WS₂ monolayer (a), Ag nanodisks (b), and WS₂–Ag hybrid structure (c). **d–f** the corresponding time evolution of the signals in (a–c). Figure reproduced from *ACS Photon.* 2019, 6, 2832.

In the article "*Adv. Optical Mater.*, 2020, 8, 2001147", Shi et al.'s coupling system was established in a single Ag nanowire cavity, where the signal from the heterostructure did not show the opposite sign compared to that from pure WS₂ region. We speculate that this distinction compared to our work (and also to other reports such as Qihua Xiong group) may come from different light-matter interaction mechanisms between nanodisk array and single nanowire, or alternatively, due to the larger pump spot than the nanowire diameter, which results in a mixed response from pure WS₂ and WS₂-Ag nanowire heterostructure.

Comment 4. In the rate equation S1 representing the whole coupled system, from my understanding from "*American Journal of Physics* 78, 1199-1202 (2010)", all the terms presenting the different coupling strengths Ω should be multiplied by the displacement x , not its derivative.

Response 4: In fact, there are two forms of rate equations concerning different definitions of coupling strength. In the mentioned literature "*American Journal of Physics* 78, 1199-1202 (2010)", the coupling strength (κ) has a dimension similar to the spring constant, and both of them are multiplied by the displacement x . While in our definition, the coupling strength (Ω) has a dimension

similar to the damping constant (or linewidth, γ), so both of them are multiplied by the rate \dot{x} rather than the displacement x . Nevertheless, these two definitions are equivalent in solving the eigenvalues of the coupling system.

Taking the two-coupling model without damping (γ) as an example, if the coupling strength κ definition is used, the Rabi splitting obtained under resonance in "American Journal of Physics 78, 1199-1202 (2010)" is

$$\omega_+ - \omega_- = \frac{\kappa}{m\omega} \quad (\text{R2})$$

While if Ω definition is used, let's start from scratch to deduce the Rabi splitting. The rate equation is

$$\begin{aligned} \ddot{x}_{\text{pla}} + \omega_{\text{pla}}^2 x_{\text{pla}} + \Omega_X \dot{x}_X &= F_0 e^{-i\omega t} \\ \ddot{x}_X + \omega_X^2 x_X - \Omega_X \dot{x}_{\text{pla}} &= 0 \end{aligned} \quad (\text{R3})$$

Neglecting the driving force $F_0 e^{-i\omega t}$ when discussing the intrinsic vibration frequency, and the rate equation can be converted into a matrix form with a tentative solution $x(t) = x_0 e^{-i\omega t}$.

$$\begin{bmatrix} \omega_{\text{pla}}^2 - \omega^2 & \Omega\omega \\ \Omega\omega & \omega_X^2 - \omega^2 \end{bmatrix} \begin{bmatrix} x_{\text{pla}} \\ x_X \end{bmatrix} = 0 \quad (\text{R4})$$

Under near resonance approximation $(\omega - \omega_{\text{pla}}) \ll \omega$, $(\omega - \omega_X) \ll \omega$, the matrix can be simplified as

$$\begin{bmatrix} \omega_{\text{pla}} - \omega & \Omega/2 \\ \Omega/2 & \omega_X - \omega \end{bmatrix} \begin{bmatrix} x_{\text{pla}} \\ x_X \end{bmatrix} = 0 \quad (\text{R5})$$

For a nontrivial solution, the matrix determinant should be zero, i.e. $|H - I\omega| = 0$, where I is the unit matrix and H is the Hamiltonian.

$$H = \begin{bmatrix} \omega_{\text{pla}} & \Omega/2 \\ \Omega/2 & \omega_X \end{bmatrix} \quad (\text{R6})$$

Thus the eigenfrequency can be solved as $\omega_{\pm} = \frac{1}{2} \left[\omega_{\text{pla}} + \omega_X \pm \sqrt{\Omega^2 + (\omega_{\text{pla}} - \omega_X)^2} \right]$. Under complete resonance where $\omega_{\text{pla}} = \omega_X$, the Rabi splitting is

$$\omega_+ - \omega_- = \Omega \quad (\text{R7})$$

Starting from different definitions of coupling strength (κ or Ω), two forms of Rabi splitting (that is, $\frac{\kappa}{m\omega}$ and Ω) are obtained, which should be essentially equivalent. Additionally, $\frac{\kappa}{m}$ has

a dimension of squared energy (ω^2), rendering $\frac{\kappa}{m\omega}$ an energy dimension, which is in line with Ω .

In summary, taking Ω definition is somewhat convenient (Kamenetskii E, Sadreev A, Miroschnichenko A. Fano resonances in optics and microwaves. Berlin: Springer, 2018), as the energy dimension of Ω can be intuitively compared with the linewidth (loss), serving as the coupling criterion for the hybrid system.

Comment 5. Finally, in Supplementary Figure 18, it is not clear what the step size of the time delay setup is, since the dense set of data points shown, but a time resolution of ~ 100 fs is declared in the Method section (3. Nonlinear transient absorption ($\Delta R/R_0$) spectrum). From that point of view, some more details about the experimental setup, accompanied by a sketch, would be really useful to guide the reader through the manuscript.

Response 5:

We are very grateful for the reviewer's feedback. The step size of 50 fs. The accuracy of this step size depends on the minimum displacement of the delay line that controls the time delay between the pump and probe. As for the time resolution of the pump-probe setup, after careful inspection and re-measurement, we have confirmed that the time resolution (full width at half maximum (FWHM)) of the system is $\tau=190\pm 20$ fs, rather than the claimed ~ 100 fs in the manuscript, due to the incorrect use of half width at half maximum (HWHM) instead of FWHM in calculation.

Specifically, there are two methods to determine the time resolution of the pump-probe system. The first one is to fit the experimentally measured ultrafast relaxation curve with an exponential decay convoluting with an instrument response function (IRF). As shown in Fig. R7 (data from Supplementary Figure 16), the ultrafast relaxation curve $g(t)$ of WS_2 exciton can be fitted with

$$g(t) = f(t) * h(t) \quad (R8)$$

where $f(t) = f_0 e^{-t/\tau_1}$ is the actual exciton relaxation, τ_1 is the lifetime, and the Gaussian function

$h(t) = h_0 e^{-\frac{t^2}{2w^2}}$ is the IRF of our setup (here, we ignored the excitation process of exciton, since it is much shorter than the time resolution). Then the resolution τ of the instrument is defined as the FWHM of the IRF, with

$$\tau = 2\sqrt{2 \ln 2} w \quad (R9)$$

By deconvolutional the relaxation curve from Fig. R7, $w=80\pm 10$ fs can be obtained, thus

$$\tau = 2\sqrt{2 \ln 2} w = 190 \pm 20 \text{ fs} \quad (R10)$$

Fig. R7 Normalized relaxation dynamics of the neutral exciton (X).

Although the above method provides actual time resolution, we can still use the second method for further verification. Firstly, the femtosecond laser from Light Conversion Ltd. (LC) yields a laser output that has reached almost Fourier transform limit, and thus the pulse width of the pump can be theoretically estimated from the spectrum of the laser, as shown in Fig. 8. For 2.19 eV pumping (the most pump energy we used in the main text), the LC software provides a pump pulse width with $\tau_{\text{pump}}=141$ fs. On the other hand, due to the use of supercontinuum white light for the probe pulse, whose width should be smaller than the pump when the chirp is corrected (as we measured the supercontinuum probe using a spectrometer with a spectral resolution of ~ 1 nm, the chirp of the probe can be corrected, this strategy is commonly used in broadband pump-probe system). Here, as a conservative estimate, it can be assumed that the probe width is equal to the pump width, i.e. $\tau_{\text{probe}}=141$ fs. Therefore, the time resolution of the pump-probe system is

$$\tau = \sqrt{\tau_{\text{pump}}^2 + \tau_{\text{probe}}^2} \approx 200 \text{ fs} \quad (\text{R11})$$

this value is completely consistent with the result obtained from fitting the IRF curve above, again confirming that the system's time resolution is ~ 200 fs.

Fig. R8 Pulse width measurement from LC software. After the pulse correction, the value $\tau_{\text{pump}}=141$

fs.

Although the time resolution of the system, or more accurately, the FWHM of the instrument response function is $\tau = 190 \pm 20 \text{ fs}$, it does not mean that only transient processes with lifetimes larger than 200 fs can be distinguished. In fact, by deconvolution fitting, we can analyze the ultrafast process several times faster than τ , as shown in Fig. R9. Generally, according to the monograph (Becker W., *Advanced Time-Correlated Single Photon Counting Techniques*, 2005, 263-346), the fastest detectable process of the system equals to τ/n , where $1 < n < 10$. Under ideal conditions of a single-exponential decay and a high signal-to-noise ratio, n can reach up to 10. Therefore, in our system, under a limited signal-to-noise ratio, we consider $n = 2$ to be a relatively conservative estimate, that is, a fastest process of $\sim 100 \text{ fs}$ can be distinguished in the system.

Fig. R9 IRF and ultrafast time-resolved process of the pump-probe system. The shaded area is the instrument response function (IRF, FWHM = 190 fs), which is fitted from the exciton relaxation curve (Fig. R7) of WS_2 , while other curves represent the relaxation dynamics with different lifetimes τ_1 .

Additionally, the schematic diagram of the experimental setup is shown in Fig. R10, which is also added in the new supplementary information (Supplementary Fig. 19)

Fig. R10 sketch of the pump-probe experimental setup. OPA: optical parametric amplifier. NA: numerical aperture.

Revision 5: We have corrected the words “with a time resolution of $\sim 200 \text{ fs}$ ” in Methods. Then, the panel of Fig. R10 is added to Supplementary Information as **Supplementary Figure 19**.

Comment 6. In Figure 1c of the main text, label at least top and bottom curves, referring to E_{pla} of Figure 1d, as done in Figure 2b with temperature.

Response 6: Thanks for the suggestion! We have added the label into Fig. 1c. For Fig. 2b, as we mentioned in the main text, one zero-detuned sample is measured.

Revision 6:

Fig. R11 Revision for Fig. 1.

Comment 7. In Figure 3a of the main text, label at least top and bottom curves, referring to the total polariton density n_{tot} .

Response 7: Thanks for the suggestion. We have added labels for the increasing densities in Fig. 3a, with the corresponding color spotted in Fig. 3b.

Revision 7:

Fig. R12 Revision for Fig. 3.

Comment 8. In Figure 3g of the main text, the circle in the trap states should be red, since electrons populate those levels. Of course, same thing applies for the corresponding Supplementary Figure 4a, where the three sub-figures should be labelled.

Response 8: Thanks for the suggestion. We mistakenly used the opposite color for the circles in the trap states and have now adjusted it.

Revision 8:

Fig. R13 Revision for Fig. 3g and Supplementary Figure 4a.

Comment 9. Lines 76 – 78 in Supplementary Information related to the energy of the absorbed photons being resonant and then smaller than that of XX-. The sentences here are a little bit misleading, so it would be better to rephrase and clarify the XX- formation process.

Response 9: Thanks for the suggestion. We have revised the relevant expression in the manuscript.

Revision 9: As the following sentence, from “Generally, the XX- absorption process can be simply understood as a monolayer semiconductor, assisted by some extrinsic state, absorbing two photons one

by one (with energy resonant with XX^- emission), and transitioning from ground state to XX^- excited state. Since the energy of the XX^- is larger than the two absorbed photons, with an energy gap of $\Delta = \hbar\omega_{X^-} - \hbar\omega_{XX^-} = 24 \text{ meV}$, energy conservation requires an energetically extrinsic state to supplement this gap.” to “Generally, the formation process of XX^- can be understood as the successive absorption of two identical photons (with energy resonant with the XX^- emission), and then the transition from the ground state to the XX^- excited state with the assistance of some extrinsic states. Since the energy of the XX^- is larger than the total energy of the two absorbed photons, with an energy gap of $\Delta = \hbar\omega_{X^-} - \hbar\omega_{XX^-} = 24 \text{ meV}$, and energy conservation requires an energetically extrinsic state (e.g. trap state) to supplement this gap.”

Comment 10. In Supplementary Figure 10 (left panels, for the heterostructure), it should be better to clearly explain the meaning of orange, red and blue curves in the corresponding caption.

Response 10: Thanks for the suggestion. We have supplemented the relevant statements in the caption.

Revision 10: The words are changed from “Supplementary Figure 10. Room temperature reflectance contrast, transmission and absorption spectra of the WS_2 -Ag ND heterostructure, pure Ag ND and pure monolayer WS_2 . The reflectance contrast and absorption spectra in WS_2 -Ag ND region are fitted using equation S4.” to “Supplementary Figure 10. Room temperature reflectance contrast, transmission and absorption spectra of the WS_2 -Ag ND heterostructure, pure Ag ND and pure monolayer WS_2 . The reflectance contrast and absorption spectra in WS_2 -Ag ND region are fitted using equation S4. The blue, red and orange lines in the left panels represent the fitted spectra of pure plasmon, pure polariton and hybrid system.”

Comment 11. In Supplementary Figure 16 caption, according to the legend of panel (c), the orange curve should be with the band redshift, while the blue curve should be without considering the band redshift. Legend and caption are then not consistent with each other.

Response 11: Thanks for the suggestion. We have corrected the caption for consistency,

Revision 11: The words are changed from “(c) Normalized relaxation dynamics of the neutral exciton **with** (blue, integrating from 2.04 eV to 2.10 eV, as marked in (a) and **without** (orange, X resonance peak) considering the band redshift, and the trion dynamics (gray triangle, T resonance peak) is also provided for comparison.” to “(c) Normalized relaxation dynamics of the neutral exciton **with** (orange, the original X resonance peak) and **without** (blue, integrating from 2.04 eV to 2.10 eV, as marked in (a)) the band redshift component, and the trion dynamics (gray triangle, T resonance peak) is also provided for comparison.”

Comments from Reviewer #3

Reviewer #3 (Remarks to the Author):

In the manuscript, the authors present a spectroscopic study of a monolayer WS₂ crystal coupled to a silver nanocavity. They show the emergence of various resonances in reflectance contrast measurements, which they attribute to exciton- (X), trion- (T) and charged biexciton- (XX-) polaritons. They present measurements of the nonlinearity and claim an "unimaginably high nonlinearity" arising from the XX- resonances, promising "more quantum achievement" in the future.

The topic of the article is certainly timely, as it address the important question how the nonlinearity of otherwise weakly interacting excitonic systems can be increased. Coupling to a silver particle is also an interesting idea as that could potentially translate excitonic nonlinearity into photon interactions.

However, I have several major concerns:

Comment 1. From the outset, it is not clear why the authors target the polariton response rather than the pure excitonic nonlinearity. Is the silver "cavity" only to enhance the coupling strength?

Response 1: Compared to pure excitons, polaritons have quite different application prospects (see Introduction in the manuscript). Meanwhile, due to the fact that polaritons inherit the nonlinearity of pure excitons and enhanced light-matter coupling, they exhibit similar but stronger nonlinearity (approximately ten times in the manuscript) under the same light injection conditions. In other words, polaritons can exhibit the same nonlinear interaction intensity as excitons when the excitation density is ~10 times smaller. Meanwhile, referring to our results in Fig. 3 and Supplementary Fig. 15, we simultaneously calibrated the nonlinear coefficients of X, XX⁻ and the matching polaritons in WS₂ and Ag-WS₂ heterostructure, which are close to the nonlinear dephasing and saturation. For X and XX⁻: $\alpha_X=(0.07\pm 0.01) \mu\text{eV} \mu\text{m}^2$, $\alpha_{XX^-}=(0.5\pm 0.1) \mu\text{eV} \mu\text{m}^2$, $n_{sX}=(8\pm 1)\times 10^5$ and $n_{sXX^-}=(2.0\pm 0.2)\times 10^4$, where the values of X are close to previous literature (*Nat. Commun.* 6, 8315 (2015), *Phys. Rev. B* 88, 045318 (2013)). For polariton: $0.21\pm 0.05 \mu\text{eV} \mu\text{m}^2$ and $0.36\pm 0.11 \mu\text{eV} \mu\text{m}^2$ for X and XX⁻, $n_{sX}=(3.0\pm 0.5)\times 10^5 \mu\text{m}^{-2}$, $n_{sXX^-}=(5.3\pm 0.8)\times 10^3 \mu\text{m}^{-2}$. Those two sets of values are approximate owing to the same excitonic origin.

Comment 2. The authors make strong claims about applications, in particular quantum applications of their work. However, no quantum effects are demonstrated and the exact prospects are not made clear. I think the authors should moderate those claims and instead clearly state the results that they do have.

Response 2: It can be confirmed that polariton has quantum applications, and its basic principle is based on the blockade effect, which is closely related to optical nonlinearity. Specifically, Polariton–polariton interactions can shift the two-polariton state to higher energies, if the energy shift is larger than the polariton linewidth, a photon resonant with the transition from the ground state to the one-polariton state will generate a polariton, but a second photon of the same color cannot excite the system further (*Nat. Mater.* 2019, 18, 213). This effect promises a prospect for processing the optical quantum information (*Nat. Photon.* 2009, 3, 696), associated with applications such as parametric downconversion (*Phys. Rev. Lett.* 2005, 94, 246401), squeezing (*Nat. Commun.* 2014, 5, 3260) and spin switches (*Nat. Photon.* 2010, 4, 361). However, in our plasmon-exciton system, the polariton equipped with strong nonlinearity is not suitable for quantum applications, which is also mentioned in the Discussion of the manuscript “**However, it should be highlighted that XX^- polariton in plasmonic nanocavity studied here is more dissipative than the one in dielectric microcavity in spite of the impressive nonlinearity, and thus it is not suitable to implement the regime of quantum optics (i.e., photon blockade)**”. Because the large linewidth accompanied by adverse photon loss obstructs the fundamental principles of quantum applications. Mercifully, due to the ultrafast response nature, the type and degree of nonlinearity can be greatly tuned by control engineering of exciton and photon in a plasmon-exciton system, as the applications demonstrated in *Nat. Commun.* 2019, 10, 3264; *light: Sci. & Appl.* 2022, 11, 94; *Opt. Mater. Express*, 2018, 8, 3851.

Revision 2: To avoid confusion, we have deleted the description of quantum applications, as “**and we anticipate the so-called charged biexciton polariton is versatile in microcavity systems and capable of more quantum achievement.**” to “**and we anticipate the so-called charged biexciton polariton is versatile in microcavity systems and capable of more sophisticated achievement.**”

Comment 3. Throughout the article, I find the argumentation not easy to follow. In particular, the authors give one possible explanation for their observations without backing up against literature or thoroughly evaluating alternatives. This begins with the assignment of the linear peaks as X, T and XX^- , continues with ad-hoc assumptions about the origins of the nonlinearity and finishes with the rate model in (S11).

Response 3: Although we are not sure which part of SI the reviewers have doubted, please allow us to start from the beginning to discuss the nonlinearity of the polaritons, and then how to use the coupled oscillator model to fit the steady-state spectra for quantitating the nonlinear coefficient, just as the general method used in the previous literature, and finally the discussion for the strong nonlinear coefficient of XX^- polariton.

1. Assignment of X, T, and XX^- complexes

There has been extensive literature confirming the energy differences between different types of excitons. According to previous reports, such as *Nat. Commun.* 2018, 9, 3718; *Nat. Commun.* 2019, 10, 1709; *Nano Lett.* 2021, 21, 2519; *ACS Nano* 2022, 16, 9728, etc., the two-particle state (X) has the highest PL energy, while the three-particle state (T) is 25~35 meV lower than X, and the five-particle state (XX⁻) is 48~55 meV lower than X. Comparing the measured R_c and PL spectra with the reported data, we can safely confirm the excitonic sources. This is exactly what we discussed in **Supplementary Note 2**. In addition, in this note, we also attempted to explain the photo absorption process of XX⁻, as few references have come to the formation mechanism of this five-particle state.

2. Nonlinearity of polaritons

We would like to clarify that in this article, as well as previous literature, the nonlinear issues of polariton mainly refer to the changes in energy, broadening, and coupling strength at different excitation densities, which is caused by repulsion, dephasing and saturation effects, respectively. To observe the nonlinearity, it is necessary to measure R_c spectra of the coupling system at different pump levels, and then acquire the density-dependent parameters such as energy E , broadening γ and coupling strength Ω , by fitting these spectra using the coupled oscillator model described below.

3. Coupled oscillator model and determination of nonlinear coefficient

Next, we will focus on how to apply the coupled oscillator model to fit the density-dependent R_c spectra and obtain the relevant parameters, which will be divided into two steps:

(1) Construction of coupled oscillator model

For the case where only one type of excitons (e.g. X) couples with photons (dielectric microcavities) or plasmons (metal nanocavities), as the room temperature case in this article, it is called two-coupled oscillator model, with motion equations as follows:

$$\begin{aligned} \ddot{x}_{\text{pla}} + \gamma_{\text{pla}} \dot{x}_{\text{pla}} + \omega_{\text{pla}}^2 x_{\text{pla}} + \Omega_X \dot{x}_X &= F_0 e^{-i\omega t} \\ \ddot{x}_X + \gamma_X \dot{x}_X + \omega_X^2 x_X - \Omega_X \dot{x}_{\text{pla}} &= 0 \end{aligned} \quad (\text{R12})$$

where ω_{pla} and γ_{pla} are the resonance frequency and linewidth of pure plasmons, ω_X and γ_X are the resonance frequency and linewidth of excitons, $F_0 e^{-i\omega t}$ represents the driving force of the light field on the plasmon oscillator. Here, assuming that the light field initially drives plasmons, and then excitons are activated through plasmon-exciton interaction, based on the fact that the absorption rate of Ag nanocavities is much higher than that of monolayer WS₂.

In the case of near resonance approximation, i.e. $|\omega - \omega_{\text{pla}}| \ll \omega$ and $|\omega - \omega_X| \ll \omega$, when discussing the intrinsic vibration of the two-level system, we can remove the driving force $F_0 e^{-i\omega t}$, Equ. R12 becomes homogeneous. By simple derivation, the system non-Hermitian Hamiltonian can be obtained as follows

$$H = \begin{bmatrix} \omega_{\text{pla}} - i\gamma_{\text{pla}} / 2 & \Omega_X / 2 \\ \Omega_X / 2 & \omega_X - i\gamma_X / 2 \end{bmatrix} \quad (\text{R13})$$

The diagonalization of the Hamiltonians yields the two polaritonic eigenfrequencies (upper polariton and lower polariton) and the Hopfield coefficients, which can be used to fit the anticrossing dispersions in Fig.1e and extract the Hopfield coefficients in Supplementary Figure 9.

On the other hand, by solving Equ. R12, the absorption or reflection coefficient of the polariton system can be written as

$$R_{1, \text{room}}(\omega) \propto \text{Im}(F_0 \dot{x}_{\text{pla}}) \propto \omega \text{Im} \left\{ \frac{S_X}{S_{\text{pla}} S_X - \omega^2 \Omega_X^2} \right\} \quad (\text{R14})$$

where $S_j = \omega_j^2 - \omega^2 - i\gamma_j \omega$ ($j = \text{pla}, X$). This solution describes the R_c spectrum of a polaritonic system where all the plasmon in the reservoir has a chance to couple with excitons. While in fact in our samples, it's almost impossible for plasmons generated at the upper ring to be coupled with the excitons at the bottom WS₂ layer, as the height of Ag nanodisk (30 nm) is much larger than the interaction length between the plasmon and exciton (~10 nm, *Nano Letters* 2021, 21, 2596). Therefore, the contribution of these pure plasmons to the reflection spectrum should be calculated separately, with

$$R_2(\omega) \propto \omega \text{Im} \left\{ \frac{1}{S_{\text{pla}}} \right\} \quad (\text{R15})$$

Then the total reflection or absorption coefficient at the WS₂-Ag NDs heterostructure at room temperature can be expressed as

$$R_{\text{room}}(\omega) \propto [kR_{1, \text{room}}(\omega) + (1-k)R_2(\omega)] \quad (\text{R16})$$

where $k \in (0 \sim 1)$ represents the polariton proportion, while $(1-k)$ dictates the pure plasmon proportion.

Equation R16 is used to fit the reflection/absorption spectrum of the heterostructure at room temperature, such as Supplementary Figure 10.

At low temperatures such as 4K, the plasmon can excite different types of excitons, including X, T and XX⁻, thus the above plasmon-X two-couple oscillator model changes to plasmon-X-T-XX⁻ four-couple oscillator model, with motion equations of

$$\begin{aligned} \ddot{x}_{\text{pla}} + \gamma_{\text{pla}} \dot{x}_{\text{pla}} + \omega_{\text{pla}}^2 x_{\text{pla}} + \Omega_X \dot{x}_X + \Omega_T \dot{x}_T + \Omega_{XX^-} \dot{x}_{XX^-} &= F_0 e^{-i\omega t} \\ \ddot{x}_X + \gamma_X \dot{x}_X + \omega_X^2 x_X - \Omega_X \dot{x}_{\text{pla}} &= 0 \\ \ddot{x}_T + \gamma_T \dot{x}_T + \omega_T^2 x_T - \Omega_T \dot{x}_{\text{pla}} &= 0 \\ \ddot{x}_{XX^-} + \gamma_{XX^-} \dot{x}_{XX^-} + \omega_{XX^-}^2 x_{XX^-} - \Omega_{XX^-} \dot{x}_{\text{pla}} &= 0 \end{aligned} \quad (\text{R17})$$

And a Hamiltonian of

$$H = \begin{bmatrix} \omega_{\text{pla}} - i\gamma_{\text{pla}} / 2 & \Omega_X / 2 & \Omega_T / 2 & \Omega_{\text{XX}^-} / 2 \\ \Omega_X / 2 & \omega_X - i\gamma_X / 2 & 0 & 0 \\ \Omega_T / 2 & 0 & \omega_T - i\gamma_T / 2 & 0 \\ \Omega_{\text{XX}^-} / 2 & 0 & 0 & \omega_{\text{XX}^-} - i\gamma_{\text{XX}^-} / 2 \end{bmatrix} \quad (\text{R18})$$

Similarly, the diagonalization of the Hamiltonians yields the four polaritonic eigenfrequencies (UP, MP2, MP1, LP, Fig.1d) and the Hopfield coefficients (Supplementary Figure 8). The reflection spectra of the polariton can also be modeled by solving Equ. R17, yielding

$$R_1(\omega) \propto \text{Im}(F_0 \dot{x}_{\text{pla}}) \propto \omega \text{Im} \left\{ \frac{S_X S_T S_{\text{XX}^-}}{S_{\text{pla}} S_X S_T S_{\text{XX}^-} - \omega^2 \Omega_X^2 S_T S_{\text{XX}^-} - \omega^2 \Omega_T^2 S_{\text{XX}^-} S_X - \omega^2 \Omega_{\text{XX}^-}^2 S_X S_T} \right\} \quad (\text{R19})$$

Combined with Equ. R15, the reflection of the heterostructure can be fitted by

$$R(\omega) \propto [kR_1(\omega) + (1-k)R_2(\omega)] \quad (\text{R20})$$

Equation R20 is used to fit the data in Fig. 2b (4 K and 120 K) and Fig. 3a. Here, by convention, the reflection data is presented in the form of R_c , with $R_c = (R_{\text{sample}} - R_{\text{substrate}}) / R_{\text{substrate}}$. Since the reflectance of the quartz substrate is approximately constant, i.e. 3.5%, R scales linearly with R_c , namely $R = 0.035(1+R_c)$.

By fitting the reflection spectrum, one can obtain the pump fluence-dependent E , γ and Ω . This is what we discussed in **Supplementary Note 3**. Note that so far we have only obtained the variation of the parameters (E , γ and Ω) with the pump fluence. To further estimate corresponding nonlinearity, we need to know the variation of these parameters with the polariton density. So in the next section (Supplementary Note 4) we need to calibrate the polariton densities at different pump fluence.

(2) Calibration of the X, T and XX- polariton densities

To calibrate the polariton density, we need to know the absorption coefficient of the polariton (A_{pol}). Note that the absorption coefficient (A) consists of two parts, the polariton and the pure plasmon components (A_{pla}), i.e., $A = A_{\text{pol}} + A_{\text{pla}}$, and here, we only concern the A_{pol} .

It is hard to determine A_{pol} because it varies with the angle frequency ω and can't be directly measured. Fortunately, previous report (*Nat. Photon.* 2015, 9, 466) show that for an ultrathin layer on a transparent quartz substrate (that is also our case here), the A_{pol} is approximately proportional to the reflectance contrast R_c , with

$$A_{\text{pol}}(\omega) = C \cdot R_{c\text{-pol}}(\omega) \quad (\text{R21})$$

where C is the scale factor, $R_{c\text{-pol}}(\omega)$ can be acquired by fitting the data in Fig.3a. So the question is how to determine the scale factor. We do this by simultaneously measuring the reflection R , transmission T_r and absorption $A = 1 - R - T_r$ spectra of the sample at room temperature, see Supplementary Figure

10 and Supplementary Figure 11, and finally $C \approx 0.8\%$.

Having acquired the polariton absorption coefficients, the total polariton density can be calculated using

$$n_{\text{tot}} = \int_{E_{\text{min}}}^{E_{\text{max}}} \frac{LP(\omega) A_{\text{pol}}(\omega)}{fB_{\text{pol}} \cdot \hbar\omega} d\omega \quad (\text{R22})$$

Then the polariton density of X, T and XX^- and be determined by

$$\begin{aligned} n_X &= n_{\text{tot}} \cdot \eta_X \\ n_T &= n_{\text{tot}} \cdot \eta_T \\ n_{XX^-} &= \frac{1}{2} n_{\text{tot}} \cdot \eta_{XX^-} \end{aligned} \quad (\text{R23})$$

where $\eta_{X/T/XX^-}$ is the relative oscillator strength of X/T/ XX^- acquired from pure WS_2 , with $\eta_X + \eta_T + \eta_{XX^-} = 1$, and the coefficient of 1/2 for η_{XX^-} accounts for the absorption of two photons to generate one XX^- .

Having acquired the pump fluence-dependent E_{X,T,XX^-} , γ_{X,T,XX^-} and Ω_{X,T,XX^-} , as well as the n_{X,T,XX^-} at different pump fluence, we can plot the dotted curves in Fig.3c-e. Finally, fitting these curves with the common saturation model:

$$\begin{aligned} \Omega(n) &= \Omega(0) / \sqrt{1 + n/n_s} \\ \gamma(n) &= \gamma_0 + \alpha(n) \cdot n \end{aligned} \quad (\text{R24})$$

The saturation intensity and thus the nonlinear coefficient $g = d\Omega/dn$ can be determined (Fig.3f).

These are what we discussed in **Supplementary Note 4**.

4. Discussion on the nonlinear origin of polariton

The plasmon polariton in Ag- WS_2 is essentially a coherent superposition of plasmon and exciton oscillators. Due to the negligible nonlinearity of plasmons, the nonlinearity of polariton essentially originates from exciton component. To verify this point, we discussed the nonlinearity of pure X, T and XX^- in pure WS_2 in **Supplementary Note 5**, which was also determined by measuring the R_c spectra (Supplementary Figure 15) at different pump fluence and then fitting the spectra with three Voigt peaks. Finally, the pump fluence-dependent response intensity (square root of the absorption sample), linewidth and energy shift are required, as shown in Supplementary Fig. 15b-c. Similarly, fitting them with the saturation model (Equ. R24), one can obtain α_{X,T,XX^-} and n_{sX,T,XX^-} of pure excitons, which were found to be in good agreement with the matching polaritons in the heterostructure, further confirming that the nonlinearity of polariton comes from exciton component rather than plasmon component.

Finally, we further discussed phenomenally the large nonlinearity source of XX^- —large Bohr radius. Due to the phase space-filling in momentum space, the saturation nonlinearity of polariton has the following relationship with the Bohr radius: $g \propto a_B^2$. Therefore, we further show how to determine the Bohr radius of XX^- in **Supplementary Note 6**.

The Bohr radius or cross-section of XX^- (a_{XX^-}) is roughly estimated using the kinetic theory relation as described by You et al. (*Nat. Phys.* 2015, 11, 477), where a charged biexciton is considered to be generated by the collision of an exciton and a trion. Therefore, one has

$$a_{XX^-} = \frac{\beta}{v_{XT}} \quad (R25)$$

where $v_{XT} = \sqrt{v_X^2 + v_T^2}$ is the relative velocity of the X and T, with the 2D Maxwellian velocity of

$v_X = \sqrt{\frac{\pi k_B T}{2m_X}}$ and $v_T = \sqrt{\frac{\pi k_B T}{2m_T}}$ respectively, and β represents the XX^- formation rate from

collisions of X and T, which can be determined by fitting the transient absorption kinetics of the X, T and XX^- using the following equation model

$$\begin{aligned} \frac{dn_X}{dt} &= -\gamma_X n_X - \beta n_X n_T + \gamma_{XX^-} n_{XX^-} / (1+D) \\ \frac{dn_{XX^-}}{dt} &= -\gamma_{XX^-} n_{XX^-} + \beta n_X n_T \\ n_T &= D n_X \end{aligned} \quad (R26)$$

The fitting results are shown in Supplementary Fig. 16d, providing an estimated value of $\beta = 0.8 \text{ cm}^2 \text{ s}^{-1}$ and thus a Bohr radius of $a_{XX^-} \approx 5 \text{ nm}$. Given that $g \propto a_B^2$, and $a_X \approx 1 \text{ nm}$, this suggests a XX^- saturation nonlinearity 25 times stronger than X (or saturation density 25 times smaller than X), which is comparable, but smaller than the experimental values in the hybrid polariton (~ 56) and pure exciton (~ 40) systems.

In summary, the basic model used in this paper is the coupling oscillator model, which is commonly used in previous literature. Furthermore, the Hamiltonian obtained from this classical model has the same expression as that from quantum mechanics. In the specific implementation, the minor difference between our model and previous literature is that we use a more complicated pla-X-T- XX^- four-coupled oscillator model, whereas previous literature generally involved a two- or three-coupled oscillator model. In addition, we separately considered the non-coupled plasmon reservoir generated at the upper ring of the Ag nanodisk since the nanodisk height is much higher than the coupling lengths between plasmons and excitons.

Comment 4. To determine nonlinearities, the polariton densities have to be determined. Why does the polariton lifetime not appear in S6?

Response 4: Thank you very much for the reviewer's question. Equation S6 provides an estimation of the total polariton density,

$$n_{\text{tot}} = \int_{E_{\text{min}}}^{E_{\text{max}}} \frac{LP(\omega) A_{\text{pol}}(\omega)}{f B_{\text{pol}} \cdot \hbar \omega} d\omega \quad (R27)$$

where $L \approx 70\%$ accounts for the excitation efficiency of the system, $A_{\text{pol}}(\omega)$ is the absorption coefficient of the polariton portion in the heterostructure, which can be obtained from Supplementary Note 4.1. $P(\omega)$ is the intensity profile of the incident supercontinuum femtosecond laser, indicating the probe fluence within per unit energy width near the frequency ω , $f=100$ kHz is the repetition rate.

Therefore, $\frac{LP(\omega)A_{\text{pol}}(\omega)d\omega}{f}$ represents the energy absorbed by the sample and formed into polaritons in a detection pulse within the energy width of $d\omega$ near the frequency of ω .

After dividing by the single photon energy $\hbar\omega$, $\frac{LP(\omega)A_{\text{pol}}(\omega)d\omega}{f\hbar\omega}$ represents the absorbed number of photons in a detection pulse within the energy width of $d\omega$ near the frequency of ω , which is approximately equal to the generated polariton number.

The number of polaritons, divided by the effective polariton area B_{pol} , is equal to the initial density of injected polaritons.

From this, equation S6 calculates the initial density of polaritons, without considering the subsequent processes such as the polariton decoupling into incoherent excitons and plasmons, especially during the excitation process within the pump duration. The polariton decay process can be ignored due to the ultrafast lifetime of the coherent coupling hybrid state of polaritons, generally tens of femtoseconds, which is much faster than the subsequent process. Therefore, the injecting initial polariton density can be safely used to calculate the polariton nonlinearity, as has been done extensively in previous studies (*Nat. Commun.* 2020,11, 3589; *Nature*, 2021, 591, 61).

Comment 5. The presented theory often relies on crude estimates, which may be sufficient for a qualitative understanding but (probably) do not support some of the conclusions the authors draw. For example: "Given that $n_s \propto 1/a^2$, a blockade radius of 5 nm suggests a XX- saturation density 25 times smaller than X, which is comparable, but smaller than the experimental values in the hybrid polariton (~ 56) and pure exciton (~ 40) systems. This implies a stronger XX nonlinearity than expected from pure polariton blockade, possibly due to the additional contribution from the aforementioned doping depletion." This discrepancy may give some indication but the estimate " $n_s \propto 1/a^2$ " simply does not allow finer conclusions without a more detailed analysis.

Response 5: Generally speaking, it is not easy to accurately and quantitatively estimate the light-matter interaction in photocavity, since many factors may affect the density calculation and thus the nonlinear coefficient estimation ($g=d\Omega/dn$), such as the material quality (defect-doping level), the coupling strength and the pump-probe efficiency in the experiment. Therefore, there are significant differences in the estimation of polariton nonlinearity in previous reports. Such as for the saturation

nonlinearity of neutral exciton (X) polaritons, g varies within a wide range of $10^{-3}\sim 10^1 \mu\text{eV}\cdot\mu\text{m}^2$ (*Nat. Nanotech.* 2018, 13, 906; *Nat. Commun.* 2020, 11, 3589; *Phy. Rev. Lett.* 2021, 126, 167401; *Nat. Commun.* 2021, 12, 2269). For comparison, the estimated g in this work is in the range of $10^2\sim 10^1 \mu\text{eV}\cdot\mu\text{m}^2$, depending on pump fluence, which is in good agreement with the previous reports and proves the rationality of the model estimation.

However, the model in our work is, after all, a commonly used phenomenological model, and does not involve deeper mechanisms, such as quantum coherence, making it difficult to obtain a more detailed conclusion.

In fact, the doping depletion phenomenon may efficiently enhance the nonlinearity of charged quasiparticle polaritons, which leads to a $\sim 10^2$ times enhancement of nonlinearity for T polariton compared to X polariton (*Nat. Commun.* 2020, 11, 3589). Similarly, the depletion effect may also exist here, but it doesn't seem to play an important role, since the nonlinearity of T polariton is only ~ 1.3 times higher than that of X, possibly caused by the larger Bohr radius of T than X. Based on this, we believe that the effect of doping depletion on XX^- should not be significant, but for the sake of rigor, and according to our estimation, the nonlinearity enhancement factor (40~56 times) of XX^- polariton is indeed larger than that calculated from Bohr radius (~ 25 times). Therefore, in the original manuscript, we claim that there may be a doping depletion phenomenon.

Revision 5: In the revised manuscript, accounting for the phenomenological and cursory nature of the model, we have replaced the expression “This implies a strong XX^- nonlinearity than expected from pure polarity blockade, possibly due to the additional contribution from the aforementioned doping depletion.” with a more tactful one, as “This implies a stronger XX^- nonlinearity than expected from pure polariton blockade, possibly due to the modeling error or other factors, such as the additional contribution from the aforementioned doping depletion effect.”

Minor comments:

Comment 6. ND is not defined in the text, and neither is PB.

Response 6: Thanks for the suggestion. We have added the relevant definitions of ND and PB.

Comment 7. Regarding the identification of peaks? Have the authors performed measurements to determine the charge state of the resonances?

Response 7: The different exciton types are determined by the energy difference between these peaks and X, consistent with a large number of previous results (*Nat. Commun.* 2018, 9, 3718; *Nat. Commun.* 2019, 10, 1709; *Nano Lett.* 2021, 21, 2519; *ACS Nano*, 2022, 16, 9728). In addition, our sample was provided by Sixcarbon Technology (Shenzhen) and obtained through the CVD-growth method. Generally, this method brings an n-doped state without special treatment, as is identified

by field effect transistor (FET) measurement (*Micromachines*, 2021, 12, 1006, where the branch current of n-type is much larger than that of p-type in the bipolar FETs), while the doping degree varies with preparation conditions. And to our knowledge, there have been no reports of so-called XX^+ in monolayer WS_2 .

Comment 8. What are the error bars in 1d/e?

Response 8: Thanks for the suggestion. The error bars have been carefully placed. For Figures 1d and 1e, we identified the peak positions of LP, MP1, MP2, and UP through local Gaussian fitting. In addition, peak finding can also be achieved by calculating the first or second derivative of a smooth curve, with the results consistent with the presented data.

Revision 8:

Fig. R14 Revision for Figures 1d and e. The error bars represent 95% confidence intervals.

Comment 9. There are no error bars in 2b. Also, the theory curves fit perfectly, because new parameters (energy, gamma, Omega) are used at every T and fluence. Given so many fit parameters, the predictive value seems limited. Can their values be fixed globally (possibly within a temperature-dependent model)?

Response 9: Due to the impact of temperature on the plasmon-exciton coupling, as well as the slight changes in the optical path and beam focusing of the measurement caused by temperature rise, there will be additional differences in R_c spectra at different temperatures in addition to the change of excitonic complexes. Accounting for the numerous fitting parameters, it is not credible to globally fix the parameters at different temperatures. In Fig. 2, as shown by the green line, we only selected the 4 K and 120 K spectral curves (corresponding to the temperatures at which XX^- polariton is most prominent and just appearing in the spectrum) for fitting and qualitative analysis. What we want to show in Fig. 2 is that the XX^- can exist at a temperature up to 120 K. But notice that the hybrid XX^- polariton state is not formed at 120 K, since the fitting coupling strength is far from the strong coupling condition, due to the large linewidth of Ag plasmon. Nevertheless, if we can further improve the cavity Q factor such as using dielectric microcavity, strong coupling between photons

and XX^- might be achieved at similar temperatures.

As for the fitting of experimental data, we employ the commonly used coupling oscillator model (Equ. R19). The slight difference is that in our model, in addition to considering the polaritons generated at the Ag nanodisk edge proximal to WS_2 , we also consider the contribution of the uncoupled plasmons generated near the upper edge of the Ag nanodisk (Equ. R15), and the total response is

$$R(\omega) = R_{\text{pol}}(\omega) + R_{\text{pla}}(\omega) \propto kR_1(\omega) + (1-k)R_2(\omega) \quad (\text{R28})$$

Thus the only difference between the model here and previous literature is the consideration of the polariton proportion k . In the fitting process, k is a free variable, and its fitting result under different pump fluence is shown in Fig. R15a (also seen in Supplementary Fig. 6). We can see that the variation of k is relatively small, between 0.55 and 0.6. This is consistent with previously reported simulations of similar structures (Fig. R15b, *ACS Photon.* 2019, 6, 2832; *ACS Photon.* 2019, 6, 286), in which plasmons are found to be almost evenly distributed at the upper and lower edge rings of the Ag nanodisk, and thus the ratio k should be approximately half.

Fig. R15 a Fitted polariton proportion k that varies with polariton density n_{tot} . The error bars represent 95% confidence interval. **b** Calculated electric field intensity distribution of the Ag nanodisk on the glass substrate with the polymer superstrate, Figure reproduced from *ACS Photon.* 2019, 6, 2832.

Of course, it is also an alternative to consider a fixed k , such as $k=0.5$, and the fitting results are shown in Fig. R16. Obviously, compared to a free k , the fit for a fixed k becomes worse at low density, especially at the intermediate position near MP1 and MP2. At high density, however, there is no significant difference.

Fig. R16 Curve fitting comparison between when a released (left) and fixed (right) $k=0.5$ parameter.

The extracting coupling strengths Ω for a released and a fixed k are shown in Fig. R17. Fitting them with the saturation model (Equ. R24), the saturation densities of X and XX' polariton are found to be $n_{sX}=(3.0\pm 0.5)\times 10^5 \mu\text{m}^{-2}$ and $n_{sXX'}=(5.3\pm 0.8)\times 10^3 \mu\text{m}^{-2}$ for a released k , where n_{sX} is ~ 56 times that of $n_{sXX'}$; and $n_{sX}=(7\pm 2)\times 10^5 \mu\text{m}^{-2}$ and $n_{sXX'}=(1.1\pm 0.2)\times 10^4 \mu\text{m}^{-2}$ for a fixed k of 0.5, where n_{sX} is ~ 64 times that of $n_{sXX'}$.

Although both n_{sX} and $n_{sXX'}$ are slightly larger for $k=0.5$, (i) n_{sX} is still comparable to the theoretical value derived from the Bohr radius $1/a_{\text{BX}}^2 \sim 10^6 \mu\text{m}^{-2}$, verifying the rationality of the fitting procedure. (ii) The ratio of n_{sX} to $n_{sXX'}$ (64) is very close to that from a released k (56), indicating that whether the fitting k parameter is fixed or not has little influence on the nonlinearity enhancement estimation of XX' polariton.

Fig. R17 Fitting results of Ω comparison between when a released (left) and fixed (right) $k=0.5$ parameter.

Therefore, we confirm that from a physical perspective, it is evident that k may fluctuate within

a small range under different power irradiation. Thus in the actual fitting procedure, we regard k as a variable.

The reason why it seems that there are too many parameters in the fitting process is that the commonly used coupled oscillator model itself involves many parameters. However, these parameters are indeed necessary and basic parameters. In equation S2, we have four S variables (S_j , $j = \text{pla, X, T or XX}^-$), and each S_j involves two variables: frequency ω_j and broadening γ_j . In addition, there are three coupling strengths Ω_j ($i = \text{X, T or XX}^-$) between excitons and plasmons. Obviously, as pump fluence changes, these parameters all change, so it is unreasonable to fix any one parameter.

$$R_1(\omega) \propto \text{Im}(F_0 \dot{x}_{\text{pla}}) \propto \omega \text{Im} \left\{ \frac{S_{\text{X}} S_{\text{T}} S_{\text{XX}^-}}{S_{\text{pla}} S_{\text{X}} S_{\text{T}} S_{\text{XX}^-} - \omega^2 \Omega_{\text{X}}^2 S_{\text{T}} S_{\text{XX}^-} - \omega^2 \Omega_{\text{T}}^2 S_{\text{XX}^-} S_{\text{X}} - \omega^2 \Omega_{\text{XX}^-}^2 S_{\text{X}} S_{\text{T}}} \right\} \quad (\text{R19})$$

where $S_j = \omega_j^2 - \omega^2 - i\gamma_j\omega$ ($j = \text{pla, X, T or XX}^-$).

Such coupled oscillator models have been widely used in previous literature, in which the issue of excessive parameters also exists, e.g. *Nature*, 2021, 591, 61 and *Nano Lett.* 2018, 18, 1777.

Comment 10. "Note that Ω_{X} at 293 K is in excellent agreement with $\sqrt{(\Omega_{\text{X}}^2 + \Omega_{\text{T}}^2 + \Omega_{\text{XX}^-}^2)}$ = 138 meV at 4 K [...]" What does this mean?

Response 10: Thanks very much for the reviewer's question. The spectral signature of strong coupling is the splitting of the excitonic absorption bands into several polaritonic new bands (e.g. LP, MP1, MP2, UP). The energy splitting of these states is governed by the coupling strength (Ω), which depends on the electromagnetic field per photon E of the optical mode, the transition dipole moment μ of the exciton, and the number of excitons N within the volume of the optical mode, as $\Omega = 2\sqrt{N} E \cdot \mu$ (*ACS Photonics* 2019, 6, 286).

When the resonant cavity plasmon interacts with WS_2 excitons, the total coupling strength depends on the total number of excitons N that can couple with plasmons within the optical mode volume, with $\Omega \propto \sqrt{N}$. During refrigeration, if the ability of light excitation to produce exciton remains unchanged, the total N is constant. At room temperature, excitons are mainly composed of X, while at low temperatures, the excitonic complexes emerge as X, T, XX^- . Thus, the agreement between Ω_{X} at 293 K and $\sqrt{\Omega_{\text{X}}^2 + \Omega_{\text{T}}^2 + \Omega_{\text{XX}^-}^2}$ = 138 meV at 4 K indicates an entire distribution of the oscillator strength between the three exciton species during refrigeration. A similar conclusion can also be seen in Zhang et al. (*Nature* 2021, 591, 61), in which the coupling strength of monolayer MoSe_2 A excitons is mostly allocated to two types of Moire excitons.

Revision 10: In the original manuscript, we overlooked the premise of this conclusion, as $\Omega \propto \sqrt{N}$, which may reduce readability. Therefore, in the revised manuscript, we have made the

following statement modification. From “Note that Ω_X at 293 K is in excellent agreement with $\sqrt{\Omega_X^2 + \Omega_T^2 + \Omega_{XX}^2} = 138$ meV at 4 K, indicating an entire distribution of the oscillator strength between the three exciton species during refrigeration.” to “Note that Ω_X at 293 K is in excellent agreement with $\sqrt{\Omega_X^2 + \Omega_T^2 + \Omega_{XX}^2} = 138$ meV at 4 K. Given that the coupling strength (Ω) depends on the number of excitons within the plasmonic volume (N), i.e., $\Omega \propto \sqrt{N}$, this consistency indicates an entire distribution of the oscillator strength between the three exciton species during refrigeration.” with the added Reference 45.

Reviewers' Comments:

Reviewer #1:

Remarks to the Author:

Dear authors,

All my questions were successfully addressed. After thoroughly reading the revised manuscript and answers to the other referees' questions, I am happy to recommend publishing the manuscript titled "Charged biexciton polaritons sustaining strong nonlinearity in 2D semiconductor-based nanocavities" in Nature Communications.

Reviewer #2:

Remarks to the Author:

The authors have made the changes to the current version of the manuscript they were requested after the previous round of revision, according to reviewers' comments. The authors have addressed the main concerns raised by the reviewers in their rebuttal, also responding in an exhaustive way to the comments, clarifying some of the statements from both a theoretical and an experimental perspective.

Overall, this revised version can be considered suitable for publication, but a series of minor technical issues should be addressed first. Essentially, extreme care should be devoted in editing the text to make reading easier, since a few grammatical mistakes and typos plagued the manuscript.

As an example, we should read "intermediate coupling conditions are also met" instead of "intermediate coupling condition is also met".

Or, to illustrate another case, I assume that "the study of biexciton-polaritons" remains "elusive" or "excluded", not "exclusive".

In Fig. 3 caption, it should be "vertical dashed line". This kind of typo is present throughout the whole manuscript. And so on and so forth.

A well-written text would make readers' task more pleasant, allowing them to better understand the authors' research without making any further effort to interpret the manuscript.

In conclusion, excluding these cosmetic (but still important) corrections, I reckon this manuscript to represent a novelty in the field of light-matter coupling and to be scientifically accurate enough to be publishable in a journal such as Nature Communications.

Reviewer #3:

Remarks to the Author:

I thank authors for their detailed explanations. Although there remain a few points regarding the interpretation of the data that I find not fully convincing, I expect the results are of sufficient interest to the community to warrant publication. I list my remaining concerns here for the authors to consider, here or in subsequent work:

1. Relating to Response 1 in the rebuttal letter: How can the polariton nonlinearity be stronger than the excitonic nonlinearity from which it originates through mixing with a photon? The Hopfield coefficient relating the two quantities is always smaller than 1.
2. Relating to Response 10 in the rebuttal letter: There should be a sum rule of the oscillator strength of the different polariton types. However, the number N of excitons in the mode volume should not be an independent parameter. Rather, the oscillator should be an intrinsic quality of the material, while the number/density of exciton depends on the excitation conditions. Since the result is obtained in the linear regime, it should be independent of N .

Comments from Reviewer #2

Comment 1.

Overall, this revised version can be considered suitable for publication, but a series of minor technical issues should be addressed first. Essentially, extreme care should be devoted in editing the text to make reading easier, since a few grammatical mistakes and typos plagued the manuscript.

As an example, we should read “intermediate coupling conditions are also met” instead of “intermediate coupling condition is also met”.

Or, to illustrate another case, I assume that “the study of biexciton-polaritons” remains “elusive” or “excluded”, not “exclusive”.

In Fig. 3 caption, it should be “vertical dashed line”. This kind of typo is present throughout the whole manuscript. And so on and so forth.

Response1: We are very grateful for the reviewer's feedback that we have used to improve the quality of our manuscript. Besides the three points mentioned above, we have carefully corrected all the typos throughout the whole manuscript, with changes to the manuscript are given in colored regions.

Comments from Reviewer #3

Comment 2.

I thank authors for their detailed explanations. Although there remain a few points regarding the interpretation of the data that I find not fully convincing, I expect the results are of sufficient interest to the community to warrant publication. I list my remaining concerns here for the authors to consider, here or in subsequent work:

1. Relating to Response 1 in the rebuttal letter: How can the polariton nonlinearity be stronger than the excitonic nonlinearity from which it originates through mixing with a photon? The Hopfield coefficient relating the two quantities is always smaller than 1.

Response2: We appreciate for the reviewer's question. In fact, the interaction strength between one single polariton and other particles is weakened relative to the exciton, because the photon-exciton hybridization introduces the non-interaction photon component (where the Hopfield coefficient relates the two quantities is always smaller than 1). However, on the other hand, due to the existence of microcavities (where the high Q factor causes sustained light-matter interaction) or nanocavities (where light-matter interaction is squeezed into the sub-wavelength extent), the overall polariton nonlinearity from heterostructure is enhanced than that of excitons from pure material,

under the same light excitation conditions. Our experiment results tell the enhancement factor is ~ 10 .

Comment 3.

2. Relating to Response 10 in the rebuttal letter: There should be a sum rule of the oscillator strength of the different polariton types. However, the number N of excitons in the mode volume should not be an independent parameter. Rather, the oscillator should be an intrinsic quality of the material, while the number/density of exciton depends on the excitation conditions. Since the result is obtained in the linear regime, it should be independent of N .

Response3: Thanks very much for the reviewer's concern. Here, our inaccurate expression brings the misunderstanding to readers. As a matter of fact, the energy splitting of polariton states is $\Omega = 2\sqrt{N}E\mu = 2E\sqrt{f}$, where E is the electromagnetic field of per photon, N is the number of exciton states in the material that enables photon absorption and excitation (determined by the intrinsic property of the material), rather than the number of excitons excited by light (determined by the excitation power), and $f = N\mu^2$ represents the total oscillator strength. During the refrigeration process, with little disturbance from impurity state and high-order excited state, the total oscillator strength remains approximately unchanged. Therefore, Ω_x at 293K and $\sqrt{\Omega_x^2 + \Omega_t^2 + \Omega_{xx}^2}$ at 4K should be in excellent agreement.

Revision 3: In accordance with the reviewer's comments, but also to avoid misunderstanding by readers, we have revised the following sentence, from “Note that Ω_x at 293 K is in excellent agreement with $\sqrt{\Omega_x^2 + \Omega_t^2 + \Omega_{xx}^2} = 138$ meV at 4 K. Given that the coupling strength (Ω) depends on the number of excitons within the plasmonic volume (N), i.e., $\Omega \propto \sqrt{N}$, this consistency indicates an entire distribution of the oscillator strength between the three exciton species during refrigeration.” to “Note that Ω_x at 293 K is in excellent agreement with $\sqrt{\Omega_x^2 + \Omega_t^2 + \Omega_{xx}^2} = 138$ meV at 4 K. Given that the coupling strength (Ω) depends on the oscillator strength (f) within the plasmonic volume, i.e., $\Omega \propto \sqrt{f}$, this consistency indicates an entire distribution of the f between the three exciton species during refrigeration.”